# Empirical stream thermal sensitivities cluster on the landscape according to geology and climate

Lillian M. McGill[1], E. Ashley Steel[2], Aimee H. Fullerton[3]

[1]Center for Quantitative Sciences, University of Washington, Seattle, WA 98105, USA, ORCID ID: 0000-0003-2722-2917
[2]School of Aquatic and Fishery Sciences, University of Washington, Seattle, WA 98105, USA, ORCID ID: 0000-0001-5091-276X
[3]Northwest Fisheries Science Center, National Oceanic and Atmospheric Administration, 2725 Montlake Blvd. East, Seattle, WA 98112, USA, ORCID 0000-0002-5581-3434

*Correspondence to*: Lillian M. McGill (lmcgill@uw.edu)

**Abstract**

Climate change is modifying river temperature regimes across the world. To apply management interventions in an effective and efficient fashion, it is critical to both understand the underlying processes causing stream warming and identify the streams most and least sensitive to environmental change. Empirical stream thermal sensitivity, defined as the change in water temperature with a single degree change in air temperature, is a useful tool to characterize historical stream temperature conditions and to predict how streams might respond to future climate warming. We measured air and stream temperature across the Snoqualmie and Wenatchee basins, Washington during hydrologic years 2015-2021. We used ordinary least squares regression to calculate seasonal summary metrics of thermal sensitivity and time-varying coefficient models to derive continuous estimates of thermal sensitivity for each site. We then applied classification approaches to determine unique thermal sensitivity regimes and, further, to establish a link between environmental covariates and thermal sensitivity regime. We found a diversity of thermal sensitivity responses across our basins that differed in both timing and magnitude of sensitivity. We also found that covariates describing underlying geology and snowmelt were the most important in differentiating clusters. Our findings and our approach can be used to inform strategies for river basin restoration and conservation in the context of climate change, such as identifying climate insensitive areas of the basin that should be preserved and protected.

## 1 Introduction

Globally, river temperature regimes are shifting in response to a changing climate. As water temperature is a critical component of aquatic ecosystems, these changes will alter an essential element of the habitat of many lotic organisms (Daufresne and Boët 2007). To apply management interventions in an effective and efficient fashion, it is critical to both understand the underlying processes causing stream warming (Arismendi et al. 2014, Steel et al. 2017) and identify the streams most and least sensitive to environmental change (Parkinson et al. 2016, Pyne and Poff 2017, Jackson et al. 2018). Measures of empirical stream thermal sensitivity, defined as the change in water temperature

with a single degree change in air temperature, or the slope of the statistical relationship between air temperature and
water temperature, address both concerns.

Thermal sensitivities reflect the combined influence of both spatially and temporally varying meteorological

and hydrological factors, and a large body of literature examines hypothesized climate, landscape, and hydrogeologic
drivers of thermal sensitivity (Table 1A). Variation in solar radiation is often the most important driver of both air and
river temperature, and as a result, air and river temperatures are typically correlated (Johnson 2003, Leach et al. 2023).
Landscape features such as riparian canopy cover and topographic shading associated with steep watersheds can
reduce exposure to solar radiation, suppressing stream temperatures (Webb and Zhang 1997). Stream temperature is
also influenced by discharge through changes to thermal inertia and residence time (Meier et al. 2003) and runoff
composition where snowmelt, surface runoff, or groundwater inflow entering the stream have different temperature
signatures than the stream itself (Webb and Zhang 1997, Mohseni and Stefan 1999, Cadbury et al. 2008). Inputs from
water sources such as snowmelt and groundwater upwelling decouple air and water temperatures and result in a
decreased thermal sensitivity of water temperature to air temperature (Tague et al. 2007, Mayer 2012, Johnson et al.
2014). As a result, the relationship between air and water temperature can also be a useful diagnostic tool for
identifying putative hydrological processes for which empirical measures are often unavailable. Thermal sensitivity
has been used in the past to estimate areas of shallow and deep groundwater influence (Snyder et al. 2015, Briggs et
al. 2018) and understand the role of snowmelt in modulating river temperature (Lisi et al. 2015, Winfree et al. 2018).
Despite conceptual agreement about hypothesized drivers of thermal sensitivity, substantial uncertainty persists
regarding the relative importance of these covariates in controlling and predicting thermal sensitivity.

Empirical stream thermal sensitivity has been widely used to characterize historical stream temperature

conditions and to predict how streams might respond to future climate warming (Mohseni et al. 2003, Mantua et al.
2010). Generally, larger thermal sensitivities indicate that water temperatures are more likely to track changes in air
temperature (Isaak et al. 2016, Mauger et al. 2017, Isaak et al. 2018b). However, there are concerns about using
current-day thermal sensitivities to predict future stream temperatures, as it can be difficult to derive insights about
river response to perturbations from statistical models that rely on historical relationships that may not extrapolate
well to future conditions. For example, past studies have found that using empirical relationships for extrapolating to
future climate scenarios without accounting for underlying processes such as snowmelt, groundwater, and annual
hysteresis may provide inaccurate predictions of future stream temperatures (Leach and Moore 2019, Steel et al. 2019).
Under changing climatic conditions, the interrelations between air temperature and other processes controlling stream
temperature may not remain stable (Arismendi et al. 2014). Additionally, stream networks can exhibit patchy thermal
conditions due to spatially heterogeneous landscape attributes such as riparian shading, valley form and aspect, and
geology (Bogan et al. 2003, Benyahya et al. 2010). Large-scale models that do not incorporate fine-scale variation in
thermal sensitivity may not accurately predict thermal habitat at ecologically relevant scales. Despite these
shortcomings, thermal sensitivity remains a commonly used and straightforward tool that allows for comparison
between locations within rivers and has the potential to guide management.
There is a need to better understand how thermal sensitivities evolve throughout the year and along river
networks and to develop a clearer understanding of the relationships between derived model coefficients and important
watershed processes. Furthermore, thermal sensitivity itself can vary across time and space, rendering stationary
values insufficient to describe variability in this parameter. A clearer vision of how thermal sensitivities vary would
allow natural resource managers to understand what a single snapshot in time or space represents and could provide
insight into how river thermal sensitivity may evolve under nonstationary air temperature and precipitation regimes.
Groups of streams (clusters) that share similar patterns of thermal sensitivity will likely also share similar risk profiles.
Identification of stream clusters could help managers tailor investment in streams according to watershed-specific
influences (Mayer 2012). This study aims to answer three questions across two Pacific Northwest river basins: **1)**
What is the spatial and temporal distribution of commonly used thermal sensitivity metrics across each basin? **2)** What
are the representative thermal sensitivity regimes , how do they cluster on the landscape, and how do these clusters
differ from clusters based on air and water temperature individually? and **3)** What are the landscape or climate factors
that best predict thermal sensitivity cluster membership? Finally, we consider the statistical functionality of these
methods on river networks.
**2 Methods**
**2.1 Study Area**
The Snoqualmie River begins as three distinct forks in the Mt. Baker Snoqualmie National Forest and drains a 1,813
km$^2$ watershed on the west side of the Cascade Range, Washington (Figure 1). The three forks originate in forested
public land before converging and flowing through a mix of agricultural, residential, and commercial land use. On
one major tributary, the Tolt River, a dam and a large reservoir provide drinking water for the City of Seattle (Figure
S4). The Wenatchee River drains 3,440 $km^2$ of the eastern Cascades before flowing into the Columbia River (Figure
S5). Although land use is similar to the Snoqualmie basin, wherein the headwaters originate in forested public lands
before flowing through a mix of agricultural, residential, and commercial land use, forest density is generally lower
in the eastern Cascades.

Both the Snoqualmie and Wenatchee basins have a Mediterranean climate with dry summers and wet, mild

winters influenced by proximity to the Pacific Ocean. The climate on the east side of the Cascades is drier than that
of the west side; the average annual precipitation is 1874 mm (939 mm) and the average annual temperature is 5.7°C
(5.3°C) for the western (eastern) Cascades . However, the prevailing westerly winds, which cross the Cascades, create
temperature and precipitation gradients that vary widely across the Wenatchee basin. In both basins, precipitation
occurs predominately from October to March. The coldest month is typically January, whereas the warmest is July.
Rivers have a mixed rain-snow hydrology with substantial winter rain and spring snowmelt, although the Wenatchee
basin receives more winter precipitation as snow. Peak flow generally occurs during winter in the Snoqualmie River
and spring in the Wenatchee River (Figure 2). Geology differs across the basins. Geology of the Snoqualmie basin is
characterized by a deep glacial aquifer in the lowland portion of the watershed, whereas in the alpine area much of the
ground surface is directly underlain by bedrock that lacks significant fracture systems (Turney et al. 1995, Bethel
2004). In contrast, the Wenatchee basin's geology consists of both an aquifer within the sedimentary bedrock of the
central and lowland areas and an overlying unconsolidated alluvial and outwash aquifer located primarily in river
valley bottoms (Montgomery Water Group 2003). The Snoqualmie and Wenatchee basins both have reaches where
water temperature exceeds regulatory thresholds established for salmonids that are protected by the U.S. Endangered
Species Act (ESA). Both basins support ESA-listed Chinook Salmon (*Oncorhynchus tshawytscha*) and Steelhead
Trout (*Oncorhynchus mykiss*) and the Wenatchee basin additionally supports populations of Bull Trout (*Salvelinus*
*confluentus*) and Sockeye Salmon (*Oncorhynchus nerka*).

Water temperature loggers ($N_{Snoqualmie}$=42, $N_{Wenatchee}$=31) were installed throughout the mainstems, on major

tributaries and on a selection of minor tributaries for both the Snoqualmie and Wenatchee rivers (Figure 1). Practical
limitations forced sites to be publicly accessible, or on private property with landowner permission, and within 1 km
of a road. For this study, water temperature was recorded using HOBO TidbiT v2 (UTBI-001) loggers every hour
from October 1, 2014 through September 30, 2021 in both basins. We hereafter use North American hydrologic years
(1 October – 30 September) instead of calendar years with the year of summer data as the year of reference. Air

temperature data was recorded using HOBO Pendant (UA-002-64) loggers every hour at all water temperature monitoring sites. Air temperature was logged for subset of 11 (6) sites in the Snoqualmie (Wenatchee) basin beginning October 1, 2014, and for all sites beginning October 1, 2016 (October 1, 2018). Air loggers were placed on trees along the stream bank, as close to the stream temperature loggers as possible. The air temperature loggers were secured at approximately breast height on the north side of the trees. Solar shields were fashioned to house both water and air temperature loggers.

**2.2 Exploratory analysis of air-water correlation summary metrics**

We calculated two summary metrics to characterize the relationship between air temperature and water temperature. For each site, summary metrics were derived from linear regressions between mean daily values of air and water temperature. The slope of this relationship, the thermal sensitivity, indicates the average difference in water temperature when comparing time periods with a one-degree difference in air temperature. For example, a thermal sensitivity of 0.5 would indicate that, based on historical data, when air temperature at a site differs by 1°C, water temperature differs on average by 0.5°C (Leach and Moore 2019). The strength of this relationship ($R^2$) is an indicator of how well water temperature can be approximated by air temperature and is calculated as the Pearson correlation value between air and water temperature. Summary metrics were calculated separately for each season. Seasons were defined as fall (October, November, December), winter (January, February, March), spring (April, May, June), and summer (July, August, September).

Watersheds for each site were delineated and covariates describing the watersheds were obtained from commonly available geostatistical products (Table 2). Covariates were divided into four broad categories: basin topography (watershed area, mean watershed elevation, average stream slope, and distance upstream), land use (percent watershed forest, riparian forest, and lake area), climate (average temperature, precipitation, and percent precipitation falling as snow), and hydrogeologic (baseflow index, hydraulic conductivity, and soil depth to bedrock). Temperature, precipitation, and percent precipitation as snow were obtained from DAYMET Daily Surface Weather data (Thornton et al. 2020) and all other landscape covariates were obtained from the Stream-Catchment (StreamCat) Database (Hill et al. 2016).

A large body of literature examines landscape-level drivers of air and water temperature correlations within rivers. Therefore, we first summarized hypothesized drivers of thermal sensitivity based on previous literature and their covarying landscape variables within our basins (Table 1A). We then conducted an exploratory analysis of the

relationship between landscape covariates and thermal sensitivity to better understand patterns in our data and set up
future hypothesis testing. Due to the correlated nature of our dataset, no formal statistical tests were conducted. We
plotted summer thermal sensitivity against hypothesized drivers, including mean watershed elevation (MWE),
watershed slope, distance upstream, percent riparian forest cover, and substrate hydraulic conductivity. Loess curves
were plotted to aid in data visualization, and correlation coefficients between thermal sensitivity and each landscape
covariate were used to quantify the strength of the linear relationship.
We also explored the relationship between spring thermal sensitivity and snowmelt, defined as the change in
Snow Water Equivalent (SWE) for a given season and denoted as ΔSWE, and between summer thermal sensitivity
and mean air temperature and total precipitation. Climatic variables were obtained from gridded DAYMET data
products (Thornton, et al. 2020) and calculated for the upstream catchment of each monitoring station.
**2.3 Spatially weighted clustering of thermal sensitivity, water temperature, and air temperature**
To identify representative regimes of air-water temperature correlations, we employed a varying-coefficient linear
model to obtain continuous, daily estimates of thermal sensitivity. We then defined a spatially weighted dissimilarity
matrix for use in clustering, which quantifies the spatial correlation in thermal sensitivity time series while accounting
for the directed river network structure. We used this spatially weighted dissimilarity matrix with agglomerative
hierarchical clustering to identify groups of sites exhibiting similar patterns in thermal sensitivity over time and
compared these clusters to those generated using only water or air temperature. Details of each step are provided in
the following sections.
**2.3.1 Varying coefficient linear model for air-water relationship**
To derive a continuous thermal sensitivity metric, we fit a time-varying coefficient model (TVCM) to air and water
temperature data. The TVCM is an effective tool for exploring dynamic features of the sensitivity of water temperature
with changes in air temperature and uses a parametric linear model but with time-varying coefficients (Li et al. 2014,
2016). For a given site, we described the varying coefficient model for the air–water temperature relationship as:
$$y_t = \beta_{0,t} + x_t\beta_{1,t} + \epsilon_t, t = 1, \dots, T \tag{1}$$
Where $\beta_{0,t}$ and $\beta_{1,t}$ are varying intercept and slope coefficients. To estimate the time-varying coefficients, we adopted
an ordinary least squares kernel regression with the Nadaraya–Watson estimator, where we fit a set of weighted local
regressions with an optimally chosen window size defined by the bandwidth, $b$, and the weights given by the kernel
function (Hoover 1998, Casas and Fernandez-Casal 2019). The kernel and its bandwidth control the level of smoothing
by adjusting the weight that the neighbouring time points have on estimates at **t**. The bandwidth was set to 0.2 a priori
to ensure consistency across time series. We used the Gaussian kernel that is of the form $k(x) = \frac{1}{2}\pi\, e^{-\frac{x^2}{2}}$. The varying
intercept term represents the mean water temperature at time *t* and the varying slope term represents the local
sensitivity of water temperature to changes in air temperature at time *t*. We used the R package tvReg (Casas and
Fernandez-Casal 2021) for implementing the model.
We filtered resultant time series for site-years with > 218 days (60% of the year) and gaps of ≤ 7 days,
yielding 250 site-years from 74 sites across both the Snoqualmie and Wenatchee basins. To capture the typical range
and timing of thermal sensitivity at each site, we created a single representative time series of thermal sensitivity at
each site by calculating the mean daily thermal sensitivity for each day of the year across all years of filtered data. We
use this average annual time series for subsequent clustering analyses. To ensure that using an average annual time
series of thermal sensitivity was an appropriate choice given the structure of our data, we conducted a supplementary
analysis to assess cluster sensitivity to interannual variability (Appendix A). Measured air and water temperature and
modelled      thermal      sensitivities      can      be      visualized      at      the      following      link:
https://lmcgill.shinyapps.io/TimeVarying_AWC/.
**2.3.2 Estimating a spatially weighted dissimilarity matrix**
To quantify spatial correlation while accounting for the directed river network structure, we developed a dissimilarity
measure for time series of thermal sensitivity, water temperature, and air temperature that incorporated spatial
correlation between sites (Haggarty et al. 2015). The general form of the proposed dissimilarity measure between sites
*x* and *y* can be written as:
$$d^c_{xy} = d_{xy}\,\widehat{cov}(h_s) \tag{2}$$
where $d^c_{xy}$ is the spatially weighted dissimilarity matrix, $d_{xy}$ is the Canberra distance (Lance and Williams 1967), and
$\widehat{cov}(h_s)$ is a valid stream distance-based covariance matrix.
To estimate $\widehat{cov}(h_s)$, we used the tail-down model that was introduced by Ver Hoef and Peterson (2010).
Due to the complex structure of the tail-down model, it is necessary to model spatial correlation on a river network
with a covariogram. We first estimated the covariance between time series at each site using a classic formula from
Cressie (1993), which states that the estimated covariance between sites *x* and *y* is given by
$$\qquad \widehat{cov}(x,y) = \sum_{t=1}^{T} \frac{\{x_t - \bar{x}\}\{y_t - \bar{y}\}}{T} \qquad\qquad (3)$$
where $x_t$ and $y_t$ are the values of the variable (thermal sensitivity, water temperature, or air temperature) at sites $x$ and
$y$ at time $t$ and $T$ is the total number of discrete times. This results in a single value which summarizes the covariance
between the time series at the two sites over the period of interest. We then plotted these point summaries of the
covariance between pairs of curves against lags (measured as stream distance) to obtain an empirical stream distance-
based covariogram. We fit an exponential covariance function to this empirical covariogram and evaluated the model
at relevant distances to obtain an estimated stream distance-based covariance matrix $\widehat{cov}(h_s)$. We used this new
covariance matrix to weight the Canberra distance matrix as shown in Equation 2. The final spatially weighted
dissimilarity matrix, $d_{xy}^c$, was then used in clustering analyses.

### 2.3.3 Agglomerative hierarchical clustering

We used agglomerative hierarchical clustering (AHC) to identify groups of sites where the patterns in thermal
sensitivity, water temperature, and air temperature were similar over time using the hclust function in R (R Core Team
2020). AHC is a common clustering method (Olden et al. 2012, Maheu et al. 2016, Savoy et al. 2019, Isaak et al.
2020) where each time series starts in its own cluster, and the hierarchy is built by repeatedly merging pairs of similar
clusters separated by the shortest distance (i.e., measured as the similarity between individual times series) until all
points are contained in a single cluster. To decide which clusters are merged in every iteration, AHC uses a dissimilarly
metric ($d_{xy}^c$, derived in Equation 2) and a linkage criterion. We used Ward's minimum variance linkage method for
clustering, where the distance between two clusters is computed as the increase in the sum of squared differences after
combining two clusters into a single cluster. The shortest of these links (minimum increase in the sum of squared
differences) that remains at any step causes the fusion of the two clusters whose elements are involved.

A difficulty associated with cluster analysis is determining the most appropriate number of clusters given the

data because no a priori optimal number of clusters exists. Clusters resulting from alternative choices can be evaluated
through internal cluster validity indices (CVI); there are a variety of CVIs, most of which combine within cluster
cohesion (intra-cluster variance) or between cluster separation (inter-cluster variance) to compute a quality measure.
There is no universally best CVI (Arbelaitz et al. 2013), therefore we calculated a suite of five CVIs, including the
Silhouette, Gap, Davies–Bouldin, Calinski–Harabasz, and generalized Dunn indices, using the NbClust R package
(Charrad et al. 2014). A final number of clusters was determined by a majority rules approach based on the optimal
number of clusters suggested by each index (Table S2).
To determine whether clusters assignment were stable, or preserved under a perturbed dataset similar to the
original and therefore likely reflective of real differences,  we conducted a bootstrapping approach where sites were
sampled with replacement and then AHC was performed on the resampled data using the fpc R package (Hennig
2020). For each bootstrapped cluster, we assessed the similarity between each new cluster and the most similar original
cluster with the Jaccard index. The Jaccard coefficient ranges from 0 to 1. Clusters with a coefficient larger than 0.75
were considered stable, clusters with a coefficient between 0.5 and 0.75 indicate that the cluster is measuring a pattern
in the data but exact site assignment may be doubtful, and clusters with a mean Jaccard coefficient of less than 0.5
were considered unstable and may not reflect a true pattern in the data (Maheu et al. 2016, Savoy et al. 2019). We
repeated the bootstrapping procedure 10,000 times; the mean Jaccard coefficient for each cluster is reported in Table

4.

**2.3.4 Identification of environmental drivers in thermal sensitivity**
We used classification and regression trees (CART; Breiman et al. 1984) to investigate the relative importance of
hydrogeologic, climatic, landscape, and basin topography attributes for predicting each site's membership to a thermal
sensitivity cluster. CART is typically used to attempt to predict membership to clusters using environmental attributes,
and it allows the modelling of nonlinear relationships among mixed variable types and facilitates the examination of
intercorrelated variables in the final model (De'ath and Fabricius 2000, Olden et al. 2008). We took an exploratory
approach to this analysis due to our relatively small sample size ($N_{Snoqualmie} = 42$, $N_{Wenatchee} = 31$), which limited our
ability to conduct statistical tests. Therefore, we calculated variable relative importance, defined as the sum of squared
improvements at all splits determined by the predictor. These values are scaled to sum to 100 (rounded). To ensure no
single site unduly impacted CART results (Krzywinski and Altman 2017), we conducted a supplementary leave-one-
out-cross-validation analysis to ensure relative importance estimates were stable across different permutations of the
data (Figure S7). We used the R package rpart (Therneau and Atkinson 2019) for implementing the CART model.
Covariates examined are described in Table 2.

**3 Results**

**3.1 General patterns in temperature, precipitation, and thermal sensitivity**

This analysis included data from seven hydrologic years, each with differing temperature and precipitation patterns. Generally, the years spanned by our dataset were warmer than the historical average (1901-2000), with wetter than average winter and fall months and drier spring and summer months (Figure S1). For the western (eastern) Cascades, all years (2015-2021) have average annual temperatures higher than the long-term average of 8.6 °C (3 °C), although individual seasons were slightly cooler than average. The year 2015 stood out as a year with an exceptionally warm winter, low snowpack, and dry spring. Temperature and precipitation patterns in the western and eastern Cascades were generally similar, however, precipitation anomalies were typically smaller in the eastern Cascades due to the overall lower precipitation in this region (Figure 2; Figure S1).

Summary metrics describing air-water temperature relationships exhibited substantial variation across time (season and year) and space. Across all season-year combinations, thermal sensitivities ranged from 0.05 to 0.97 (mean = 0.54) in the Snoqualmie basin and from 0.06 to 0.74 (mean = 0.42) in the Wenatchee basin (Table 3). Seasonal distributions of thermal sensitivities differed. For example, fall thermal sensitivities were relatively homogeneous, with 90% of values falling between 0.47 and 0.70, whereas spring and summer thermal sensitivities exhibited a broader range of values, with 90% of values falling between 0.30 and 0.84 in spring and 0.25 and 0.78 in summer. Air temperature was generally a good predictor of water temperature, as evidenced by $R^2$ values that ranged from 0.20 to 0.99 (mean = 0.88) in the Snoqualmie basin and from 0.08 to 0.98 (mean = 0.85) in the Wenatchee basin (Table 3).

Overall, weak and inconsistent patterns emerge in summer between thermal sensitivity and landscape and climate variables (Figure 3; Table 1B). For climate variables, only ΔSWE appeared to have a linear relationship with thermal sensitivity (Figure 3). The relationship between ΔSWE and thermal sensitivity was negative and non-linear, displaying a wedge-shaped pattern wherein large snowmelt events did not reduce thermal sensitivities below 0.25 (Figure 3). For landscape variables, correlation coefficients were overall small ($|\rho| < 0.3$), indicating weak to non-existent linear relationships between landscape covariates and observed thermal sensitivity (Table 1B). A weakly negative relationship between thermal sensitivity and distance upstream was observed for both basins. Percent riparian forests and thermal sensitivity showed no relationship for either basin. The relationship between hydraulic conductivity and thermal sensitivity was weakly positive and parabolic in the Snoqualmie basin.

**3.2 Patterns of clustering for water temperatures, air temperatures, and thermal sensitivities**

Time-varying thermal sensitivities displayed periods of both high and low values within a season, which was not necessarily represented when looking only at seasonal summary metrics (Figure 4 and Figure 5). Thermal sensitivity varied alongside water and air temperature within the Snoqualmie and Wenatchee basins. Generally, thermal sensitivity rose sharply in late spring, was highest in late summer, declined slowly throughout the fall, and remained depressed through winter and early spring.

Spatially weighted AHC yielded four clusters for thermal sensitivity, with a cluster validity index (CVI) range of 2-4, and two clusters each for air (CVI range of 2-5) and water (CVI range of 2-4) temperature in the Snoqualmie basin, and five clusters for thermal sensitivity (CVI range 2-5) and two clusters each for air (CVI range of 2-3) and water (CVI range of 2-5) temperature in the Wenatchee basin (Figure 4; Figure 5; Table S2). For both basins, clusters of air and water temperature correspond closely with elevational gradients (Figure S4; Figure S5). Higher elevation sites exhibited generally lower magnitudes but similar patterns in air and water temperatures (Table 4). For example, within both basins seasonal water temperatures were synchronized, with the cluster minimum and maximum water temperatures occurring within a day of each other (Table 4). In the Snoqualmie basin, air temperature clusters were stable, with a mean Jaccard index of 0.91 for high elevation sites (Cluster 2, n=11 sites) and 0.73 for low elevation sites (Cluster 1, n=31 sites). Water temperature clusters were slightly less stable, with a mean Jaccard index of 0.65 for high elevation sites (Cluster 2, n=17 sites) and 0.89 for low elevation sites (Cluster 1, n=25 sites). Air and water temperature clusters in the Wenatchee basin were more stable than the Snoqualmie clusters. In the Wenatchee basin, air temperature clusters had a mean Jaccard index of 0.85 for high elevation sites (Cluster 2, n=25 sites) and 0.95 for low elevation sites (Cluster 1, n=6 sites), and water temperature clusters had a mean Jaccard index of 0. 86 for high elevation sites (Cluster 2, n=23 sites) and 0.73 for low elevation sites (Cluster 1, n=8 sites).

Clustering patterns for thermal sensitivity were more complex and less stable than air and water temperature clusters, particularly for the Snoqualmie basin (Figure 4; Figure 5; Table 4). In the Snoqualmie basin, Cluster 1 (n=11 sites) consisted primarily of low elevation tributaries that exhibited stable thermal sensitivities throughout the year, producing a cluster-average range of only 0.15 (Figure 4; Table 4). Cluster 2 was small (n=5 sites), and the distribution of sites within this cluster included three mainstem sites and two high elevation tributaries. Despite the large geographic distances separating sites, this cluster was highly stable with a mean Jaccard index of 0.88. Cluster 2 was characterized by a mean thermal sensitivity of 0.52 and the highest annual variability, with a cluster-average range of

0.45. Cluster 3 was large (n=15 sites) and contained sites located within the upper regions of the Snoqualmie River.
Cluster 3 had the lowest mean thermal sensitivity (mean=0.40). Lastly, Cluster 4 (n=11 sites) exhibited the lowest
stability of any cluster in the Snoqualmie basin, with a mean Jaccard index of 0.55. Sites in this cluster were mainly
situated on the mainstem Snoqualmie and its major tributaries. This cluster was distinguished by the highest mean
thermal sensitivity (mean=0.65). In the Wenatchee basin, all five thermal sensitivity clusters were relatively stable.
Clusters 1 (n=7 sites), 4 (n=8 sites), and 5 (n=8 sites) demonstrated similar seasonal patterns in thermal sensitivities,
with minimum values occurring in late Spring (water days 216, 207, 214) and maximum values occurring in late
summer (water days 324, 331, 330). These clusters also showed moderate to high stability (mean Jaccard indices of
0.79, 0.86, and 0.79). Cluster 3 (n=7 sites) exhibited the highest mean thermal sensitivity (mean=0.40) and
encompassed primarily low elevation tributaries (Peshastin and Mission Creek; Figure S5). Cluster 2 was unique in
that it consisted of a single site (Chumstick Creek) that was nearly always assigned to a unique cluster when included
in the bootstrapping procedure. The thermal sensitivity for this site was low (mean=0.29) and virtually flat throughout
the year (range = 0.07).
CART analysis indicated that basin topography and hydrogeology were the principal discriminators of
thermal sensitivity clusters. The top predictors of cluster membership (i.e., predictors with a greater than 10% increase
in mean standard error if removed from the model) were MWE and baseflow index in the Wenatchee basin and
watershed slope, MWE, and soil depth in the Snoqualmie basin (Figure 6). Variable importance distributions differed
between the Wenatchee and Snoqualmie basins, although in both basins several covariates had similar relative
importance values. Covariate distributions also varied across clusters within a basin. In the Snoqualmie basin, Cluster
1 sites were generally below a MWE of 600 meters, whereas Cluster 3 sites were generally mid-sized and high
elevation with a low baseflow index. In the Wenatchee basin, Cluster 1, 4, and 5 sites were predominately located at
high elevations with steep slopes. Cluster 4 sites exhibited a large proportion of precipitation falling as rain. Sites in
Clusters 2 and 3 were generally low elevation sites with a high baseflow index and soil depth.
**4 Discussion**
Thermal sensitivity varies throughout the year and reflects hydrologic conditions at a given time and place within a
watershed; therefore, it should not be conceptualized as a static value. Although summary metrics of thermal
sensitivity, such as average values over the summer, can still prove useful and informative, it is essential to
acknowledge the non-stationarity of the relationship between air and water temperature to obtain an accurate
understanding of how river temperature responds to changing conditions. We find that underlying geology and climate
are important controls on thermal sensitivity across two Pacific Northwest river basins, and thermal sensitivities reflect
aspects of river dynamics not redundant with water and air temperature. Overall, this study provides a framework for
using thermal sensitivity regimes to improve understanding of factors contributing to stream temperatures and will
enable managers to target mitigation and adaptation activities to work best with local conditions within a watershed.
**4.1 Patterns of thermal sensitivity clustering**
Our analysis of stream air and water temperatures supports the presence of distinct thermal sensitivity regimes,
providing an organizing framework for river research and management by identifying sites with similarities across the
network. We found that thermal sensitivity regimes reflected non-redundant aspects of river dynamics relative to air
and water temperature alone. Air temperature and water temperature clusters closely corresponded to one another and
were almost entirely determined by elevation of the temperature loggers, whereas thermal sensitivity clusters showed
more variability in annual patterns and were intermixed spatially (Figure 4; Figure 5). Previous studies within the
Pacific Northwest found that, generally, colder streams are less sensitive to air temperature fluctuations than warmer
streams (Luce et al. 2014, Kelleher et al. 2021). Air and water clustering results are consistent with previous studies
that observed broad temporal correspondence of air and river temperature dynamics with differing magnitudes of
response (Bower et al. 2004, Chu et al. 2010, Garner et al. 2014, Isaak et al. 2018a). More locally, Isaak et al. (2020)
found that across western rivers, much of the information in stream temperature records could be summarized by a
relatively limited number of distinct regime components primarily driven by differences in elevation and latitude.

Viewing thermal sensitivity as a continuous parameter adds novel insights to our understanding of river basin

functioning. Studies have highlighted the importance of annual shifts in the processes that drive heat budgets as well
as the non-stationarity of the resulting statistical relationships (Arismendi et al. 2014, Boyer et al. 2021). Our clustering
analysis overcomes these issues by using a varying coefficient model that treats thermal sensitivity as a continuous
function through time, rather than a series of discrete summary metrics, and allows clustering based on the entirety of
average annual patterns. The observed complexity in thermal sensitivity response hints at the diversity of physical
processes controlling stream temperature response and the large, clear shifts in thermal sensitivity magnitude across
the year calls into question the common practice of summarizing a river's sensitivity as a static value.  The ability to
directly observe shifts in the air-water temperature relationships also opens the possibility of using thermal sensitivity
as a diagnostic tool to examine gradual changes in the importance of drivers of water temperature, such as dynamic
changes in riparian shading or snowmelt.

**4.2 Climate controls on thermal sensitivity**

Seasonal variability of thermal sensitivity metrics was evident for our basins. Within both the Snoqualmie and
Wenatchee basins, winter thermal sensitivities were low and varied strongly with MWE (Figure 1). Observed low
thermal sensitivities in winter were likely due to the non-linear relationship between air and stream temperature at
cold temperatures when air temperatures can dip below the water temperature-freezing limit (Mohseni et al. 1998,
1999). Air temperature covaries strongly with elevation in Pacific Northwest basins, and sites that are high in the
watershed will experience a greater number of sub-freezing days, and therefore greater decoupling between air and
water temperatures. Fall thermal sensitivities were relatively homogeneous whereas spring and summer thermal
sensitivities exhibited a broader range of values. We expect thermal sensitivities to be similar during periods of heavy
precipitation, when water sources with thermal characteristics distinct from air temperature, such as groundwater and
snowmelt, contribute relatively less flow. The greater variability of responses in spring and summer indicates that the
relative magnitude of energy exchange processes controlling river temperatures are more diverse than in fall or winter
(Hrachowitz et al. 2010).
Snowmelt likely contributed to observed differences in thermal sensitivity across sites in spring and early
summer. For summary metrics, the relationship between snowmelt and spring thermal sensitivity formed a wedge-
shaped pattern, wherein sites with limited snowmelt displayed both high and low thermal sensitivity, but sites with
extensive snowmelt always display low thermal sensitivity (Figure 3). For the clustering analysis, although the
proportion of precipitation falling as snow showed limited variable importance, MWE and slope covaried closely with
snow accumulation and were among the most important predictors of cluster membership, perhaps masking a
statistical signal of snowfall (Figure 6). In both the Snoqualmie and Wenatchee basins, clusters with higher elevation,
steeper slope, and greater snowmelt within the catchment had thermal regimes that were less sensitive to changes in
air temperature during spring and early summer. Importantly, snowmelt buffering, the process wherein snowmelt-
influenced streams have lower thermal sensitivity due to a direct input of cold water and a corresponding increase in
flow rates and water depths (van Vliet et al. 2011, Siegel et al. 2022), diminishes throughout the summer. By late
summer, high elevation, snowmelt influenced sites were often more sensitive to air temperatures than their low
elevation counterparts (Figure 4; Figure 5). Sites within Cluster 4 in the Wenatchee basin were an exception to this
pattern and maintained summer thermal sensitivities that were substantially depressed relative to adjacent locations
(e.g., Clusters 1 and 5). This is likely due to snowmelt inputs within these catchments, and points to the importance
of high elevation, late-summer snowpack melt as a significant source of summer baseflow and control on water
temperatures during the months of greatest heating within these watersheds.

Numerous studies have examined the buffering impact of snowmelt on water temperature due to advective

flux from cooler meltwater entering the river. Studies in Alaskan rivers found a linear, negative relationship between
summer thermal sensitivity and snowmelt (Lisi et al. 2015, Cline et al. 2020) and a recent study in the Snoqualmie
basin found that snowmelt can reduce basin-wide peak summer temperatures, particularly at high elevation tributaries,
and the thermal impacts of melt water can persist through the summer (Yan et al. 2021). Our results suggest that
snowpack offers substantial buffering to changes in air temperature across mountain river basins, but that the largest
impacts are localized across space and time. Climate change is expected to shift snowmelt earlier and reduce snow
water resources (Barnett et al. 2005, Musselman et al. 2021). The loss of snow may result in warming in snow-
influenced systems and the subsequent homogenization of thermal conditions across basins (Winfree et al. 2018).
Homogenization of thermal conditions likely leads to important changes in ecological functions and ecosystem
services supported by lost thermal heterogeneity, such as a loss of cold-water patches for Pacific salmon (Brennan et
al. 2019).
**4.3 Hydrogeologic controls on thermal sensitivity**
Hydrogeologic characteristics shaped the relationship between air and water temperatures across the Wenatchee and
Snoqualmie basins. The inclusion of baseflow index, hydraulic conductivity, and soil depth in determining cluster
membership (Figure 6) implies the importance, and detectability, of groundwater as a key mediator of thermal
sensitivity regimes in Pacific Northwest basins. Clusters with high baseflow index, hydraulic conductivity, and soil
depth values generally had lower summer and less variable thermal sensitivities (Figure 4; Figure 5; Figure 6),
implying greater groundwater influence (Kelleher et al. 2012). Interestingly, despite the clear importance of
hydrogeologic metrics in the clustering analysis, results from summary metric exploratory analysis were mixed and,
in the Snoqualmie basin, did not align with expectations of a negative relationship between thermal sensitivity and
groundwater influence (Table 1B). Although it is possible to infer broad patterns in surface-groundwater connectivity
using datasets of interpolated geologic properties (i.e., hydraulic conductivity, soil depth) or water source (i.e.,
baseflow index), individual hydrogeologic metrics often have substantial uncertainty, do not covary perfectly, and
may be particularly unconstrained for mountain headwater streams (Wolock et al. 2004, Patton et al. 2018, Briggs et
al. 2022). Additionally, the influence of these processes can be localized and variable across space (Johnson et al.
2017) and substantially impacted by human modification. The ability to use thermal sensitivity as an empirical
measure of groundwater influence, therefore, shows great promise for understanding catchment processes and
informing management and restoration actions at ecologically relevant scales (Snyder et al. 2015). Although our
approach moves us closer to a mechanistic understanding of the relationship between thermal sensitivity and
groundwater, mixed results from our analyses emphasize the need for additional targeted studies.

An investigation of the underlying geology across the Snoqualmie and Wenatchee basins supports our

conclusion that low thermal sensitivities are indicative of groundwater inputs. The lowland portion of the Snoqualmie
watershed contains a deep, permeable, productive glacial aquifer that is presumed to be the source of summer baseflow
to much of the river (Bethel 2004, McGill et al. 2021, Turney et al. 1995). Glacial and interglacial deposits in the
valley contain several geohydrologic units with differing aquifer potential (Bethel 2004); however, most deposits can
form small but useable aquifers that could be helping to sustain baseflow in summer months (Turney et al. 1995,
Soulsby et al. 2004, Blumstock et al. 2015). Soil depth, hydraulic conductivity, and baseflow index were
correspondingly high in streams from Clusters 1 and 4 that overlay the lower portion of the watershed (Figure 6).
Thermal sensitivities reflected this pattern, wherein generally sites draining low elevation tributaries (Cluster 1) had
relatively constant thermal sensitivities throughout the year (Figure 4). Conversely, the upper portion of the
Snoqualmie basin is covered by thin soil over impermeable bedrock lacking extensive fracture networks, meaning that
rain and snowmelt are not retained in the mountains but are rapidly transmitted to the stream system (Debose and
Klungland 1964, Nelson 1971, Goldin 1973, 1992). Sites with catchments predominantly within this upland area
tended to belong to Clusters 2 and 3 and displayed high summer thermal sensitivities, perhaps indicating limited
groundwater influence.

In the Wenatchee basin, two major aquifers exist: an aquifer within the sedimentary bedrock of the central

and lowland areas and an overlying unconsolidated alluvial and outwash aquifer located primarily in river valley
bottoms across the basin (Montgomery Water Group 2003). The bedrock aquifer consists of sandstones and shales,
which tend to have moderately low permeability. Folding and faulting have caused the shale to break up or fracture
and groundwater moves preferentially within these zones of higher secondary permeability. The alluvial and outwash
aquifers, on the other hand, exhibit relatively high permeability where groundwater can move easily and are considered
the primary groundwater source (Wildrick 1979, Montgomery Water Group 2003). Cluster 2 in the Wenatchee basin,
consisting of a single site located at the mouth of Chumstick Creek (Figure S5), stands out for having a unique, nearly
flat thermal sensitivity compared to patterns at other sites (Figure 5). Covariate distributions for the clustering results
showed that Chumstick Creek has a relatively high hydraulic conductivity and baseflow index (Figure 6; Figure S8).
A transition from low to high permeability glacial material occurs near the mouth of Chumstick Creek (Montgomery
Water Group 2003), and it is possible that substantial groundwater discharge occurs near this discontinuity (Neff et
al. 2019). Similarly, sites within Cluster 3 showed low variability in thermal sensitivity and had high soil depth and
baseflow index values. Streams within this cluster are situated on top of predominantly sandstone bedrock (Frizzell
1979, Gendaszek et al. 2014).

Overall, the importance of groundwater is consistent with previous studies, which find that thermal sensitivity

decreased with increasing groundwater contribution (O'Driscoll and DeWalle 2006, Chang and Psaris 2013, Beaufort
et al. 2020, Georges et al. 2021). The degree to which groundwater decouples trends in stream and air temperature
depends on stream volume, the rate of groundwater inflow, and the depth of groundwater source. Although not
examined in this study, aquifer source and groundwater depth likely influence thermal sensitivity estimates, with
runoff sourced from deep groundwater being less variable and less sensitive in comparison to groundwater sourced
from shallow sub-surface flows (Tague et al. 2007, Johnson et al. 2021, Hare et al. 2021). Shallow groundwater
temperatures are already responding to climate change (Menberg et al. 2014). As warming continues, the summer
cooling capacity of groundwater may be reduced, limiting the availability of cold-water refugia patches sourced by
groundwater (Brewer 2013, Briggs et al. 2013).
**4.4 Landscape controls on thermal sensitivity**
Variable relationships between thermal sensitivities and landscape covariates highlight complexities in stream thermal
regimes. For example, mean channel slope was an important predictor of cluster membership for both the Snoqualmie
and Wenatchee basins, but showed a weak-to-non-existent relationship with summer thermal sensitivity summary
metrics. Steeper channel slopes and greater stream velocities limit warming in streams by decreasing the time for
equilibration with local heating conditions (Donato 2002, Webb et al. 2008, Isaak et al. 2012) and topographic shading
associated with steep watersheds can suppresses stream temperature by reducing exposure to solar radiation (Webb
and Zhang 1997). In the Wenatchee basin, the Cluster 3 site, Chumstick Creek, drains a steep canyon. This may
contribute to observed low, stable thermal sensitivities throughout the year. Additionally, watershed size and distance

upstream covary closely and displayed relatively consistent relationships with summer thermal sensitivity summary
metrics despite ranking moderately in variable importance. We expected thermal sensitivity to increase with river size;
groundwater influence should be more visible on smaller streams because the volume of water is small and the travel
time of the water from the source is short and not sufficient to equilibrate water temperature with the atmosphere
(Mohseni and Stefan 1999, Tague et al. 2007, Beaufort et al. 2016). Reduced sensitivity of headwater streams to air
temperature was observed in the Aberdeenshire Dee, Scotland (Hrachowitz et al. 2010), and River Danube, Austria
(Webb and Nobilis 2007), and small Pennsylvanian streams were shown to be less sensitive to changes in air
temperature than larger streams (Kelleher et al. 2012). However, Hilderbrand et al. (2014) found no relationship
between thermal sensitivity and watershed size in Maryland streams.

We expected landscape covariates to be important predictors of thermal sensitivity regimes, however, these
covariates were of limited importance and showed no relationship with summary metrics (Table 1B; Figure 6). Several
factors may account for this. Inherent covariation in river basins can hinder statistical efforts to identify mechanistic
links between landscape gradients and features of aquatic ecosystems (Lucero et al. 2011); land cover characteristics
may have a small impact that went undetected due to noisy observations or limited variability within our study region.
It is also possible that land cover metrics may not adequately describe the intended process. For example, the relative
unimportance of riparian shading may be due in part to our metric of shade, which was limited to riparian forest cover
and ignored topographic shading and vegetation height. Lastly, human modifications to the river that are not captured
by land cover statistics, such as channelization or the presence of dams and reservoirs, may alter thermal sensitivity
and obscure natural gradients. For example, areas of the river that are degraded and subsequently disconnected from
their floodplain may have artificially high thermal sensitivities, and the release of water from dams and reservoirs has
the potential to either warm or cool downstream temperatures, depending on dynamics of where and how impounded
water is released (Ahmad et al. 2021, Cheng et al. 2022). Future research could include covariates sinuosity or variance
of thalweg depth to better capture these effects. Untangling exact controls will require additional research.

**4.5 Assessment of statistical approach**

Collecting data on dynamic stream networks over time has inherent challenges that lead to relatively low sample sizes
and missing data as well as complex correlation structures across space and time. Our statistical approach was
designed to manage these challenges, enabling exploration of several hypotheses. These data, collected at a relatively

large number of sites in a parallel structure across two basins allow an assessment of how sensitive the statistical
approach may be to these constraints.
The time series of both air and water temperature used in this analysis have periods of missing values that
span weeks to months. Classical clustering techniques require complete datasets, limiting analyses to time series
without gaps. To overcome this issue, we calculated a single representative time series at each site that captures the
typical range and timing of thermal sensitivity. Alternative options for dealing with missing values include removing
data points that do not cover the target time period or imputing missing values by means of statistical procedures or
summary metrics (e.g., Savoy et al. 2019, Beaufort et al. 2020). However, we chose not to use these approaches in our
study due to the long and inconsistent periods of missing values across sites. We acknowledge that interannual
variability in precipitation and temperature impacts river thermal sensitivity, and average time series calculated from
differing years may exhibit differences in shape and timing for reasons outside of inherent characteristics (Appendix
A). Future studies could use novel clustering methods capable of dealing with sparse datasets, which would provide
more detailed information on clusters generated from time periods with robust values versus data scarcity (Carro-
Calvo et al. 2021). Alternatively, recent advances in space-time imputation for river basins may prove a fruitful
direction (Li et al. 2017).
Our calculation of time-varying thermal sensitives also necessitated decisions regarding what features of the
time series to preserve. Selection of the bandwidth parameter and kernel function for the time varying model will
impact estimation of thermal sensitivity and intercept. Generally, with larger bandwidth estimates or averaging periods
(e.g., daily, weekly, monthly), intercept estimates increase and thermal sensitivity estimates decrease. Decisions of
this nature should be approached carefully and with a clear question in mind. For this study, we were interested in
seasonal to annual patterns in thermal sensitivity, and thus chose a bandwidth of 0.2, resulting in a smooth seasonal
time series. Previous studies have also used regression splines to estimate the time varying relationship between air
and water temperatures (Haggarty et al. 2015). This approach smooths data and can account for missing data but may
not preserve small-scale features of interest. We chose to use absolute values of our thermal sensitivity time series, as
we cared about differences in mean thermal sensitivity as well as correlated variability. Future work could normalize
thermal sensitivity time series first to examine only patterns.
While general patterns could be detected through our analysis, the details were sensitive to exactly which
sites were sampled and included in the analysis (Figure S7).  In dynamic river systems with high spatial heterogeneity
and inherent difficulties with accessing certain areas of the network, this is always likely to be true. Our approach of
averaging across years and clustering across sites appears to manage these realities well and provide general guidance
on the river networks sampled. For example, cross validation results for CART modelling suggest that certain
variables were consistently identified as more influential for cluster prediction and that results were relatively robust
to the inclusion of individual data points (Figure S7). Strengthening the assessment of underlying drivers and controls
to provide guidance for unsampled river networks will require that similar data sets are collected across more and
more river networks. Data can then be assembled and analysed to provide more general conclusions about geologic
and climatic controls of river thermal regimes.
**4.6 Implications for management and future directions**
Classifying rivers based on thermal sensitivity could be a powerful tool when planning for global change. Our results
show that annual patterns in thermal sensitivity are diverse and mediated by underlying geology and climate across
two Pacific Northwest river basins. Climate change is decreasing snowpack in the region, resulting in earlier runoff
and extended summer baseflow (Elsner et al. 2010, Wu et al. 2012), and may decrease groundwater discharge
depending on sources and timing of recharge (Brooks et al. 2012, McGill et al. 2021). For many of our study sites,
thermal sensitives were highest in late summer during the hottest, lowest flow portion of the year. Previous studies
have found that the impact of fluctuations in discharge generally increases during dry, warm periods, when rivers have
a lower thermal capacity and are more sensitive to atmospheric warming (van Vliet et al. 2013). High thermal
sensitivity in late summer and in high elevation streams, which are typically thought to be climate refuges, is therefore
troubling for the conservation of native coldwater species such as Pacific salmon (Mantua et al. 2010; Isaak et al.
2016). Climate change will likely decrease juvenile rearing and spawning habitat quantity and quality, although it is
important to note that streams with high thermal sensitivity may still provide adequate habitat in select portions of the
year if stress-related thresholds are not exceeded (Armstrong et al. 2021).
Examining thermal sensitivity regimes improves understanding of factors contributing to stream
temperatures and may enable managers to target mitigation and adaptation activities to work best with local conditions,
thus maximizing benefits given limited resources. For example, given the importance of subsurface geology within
the Wenatchee and Snoqualmie basins, targeted actions to restore floodplain functions that recharge aquifers through
actions such as placing engineered logjams or reintroducing beavers could be prioritized (Abbe and Brooks 2013,
Pollock et al. 2014, Jordan and Fairfax 2022). Additionally, identification of particularly insensitive portions of the

river could help to better constrain areas where coldwater patches exist that may be used as refuges for coldwater fish (Snyder et al. 2020). This process-based approach will be particularly important as non-stationary relationships caused by climate change make it unreliable to use past regressions built under historical climate conditions (Boyer et al. 2021). Furthermore, as longer, more spatially extensive air and water temperature time series become available (Isaak et al. 2017), we can begin to ask questions about 1) the spatial extent of different thermal sensitivity regimes, 2) how interannual variability shifts with climate conditions and geographic context, and 3) detect changes in the external drivers of thermal sensitivities. Such insights will improve our understanding of river ecosystems while offering a suite of new tools for monitoring the impact of management decisions and climate change.

## Acknowledgements

We thank Amy Marsha, Roxana Rautu, Akida Ferguson, Shannon Claeson and the many volunteers for help collecting air and water temperature data, and Gordon Holtgrieve, Mark Scheuerell, and Christopher Jordan for suggestions that improved the manuscript. This material is based upon work supported by the National Science Foundation Graduate Research Fellowship under Grant No. DGE-1762114. Any opinion, findings, and conclusions or recommendations expressed in this material are those of the authors and do not necessarily reflect the views of the National Science Foundation.

## Author Contributions and Data Availability

All authors conceptualized the study and retrieved the data. LMM analyzed the data and prepared the manuscript with the assistance of EAS and AHF. The data that supports the findings of this study are available at https://github.com/lmcgill/AirWaterCorr/tree/master/data and can be visualized at https://lmcgill.shinyapps.io/TimeVarying_AWC/. The authors have no competing interests to declare.

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

to a warming climate. Environmental Research Letters 16:054006.


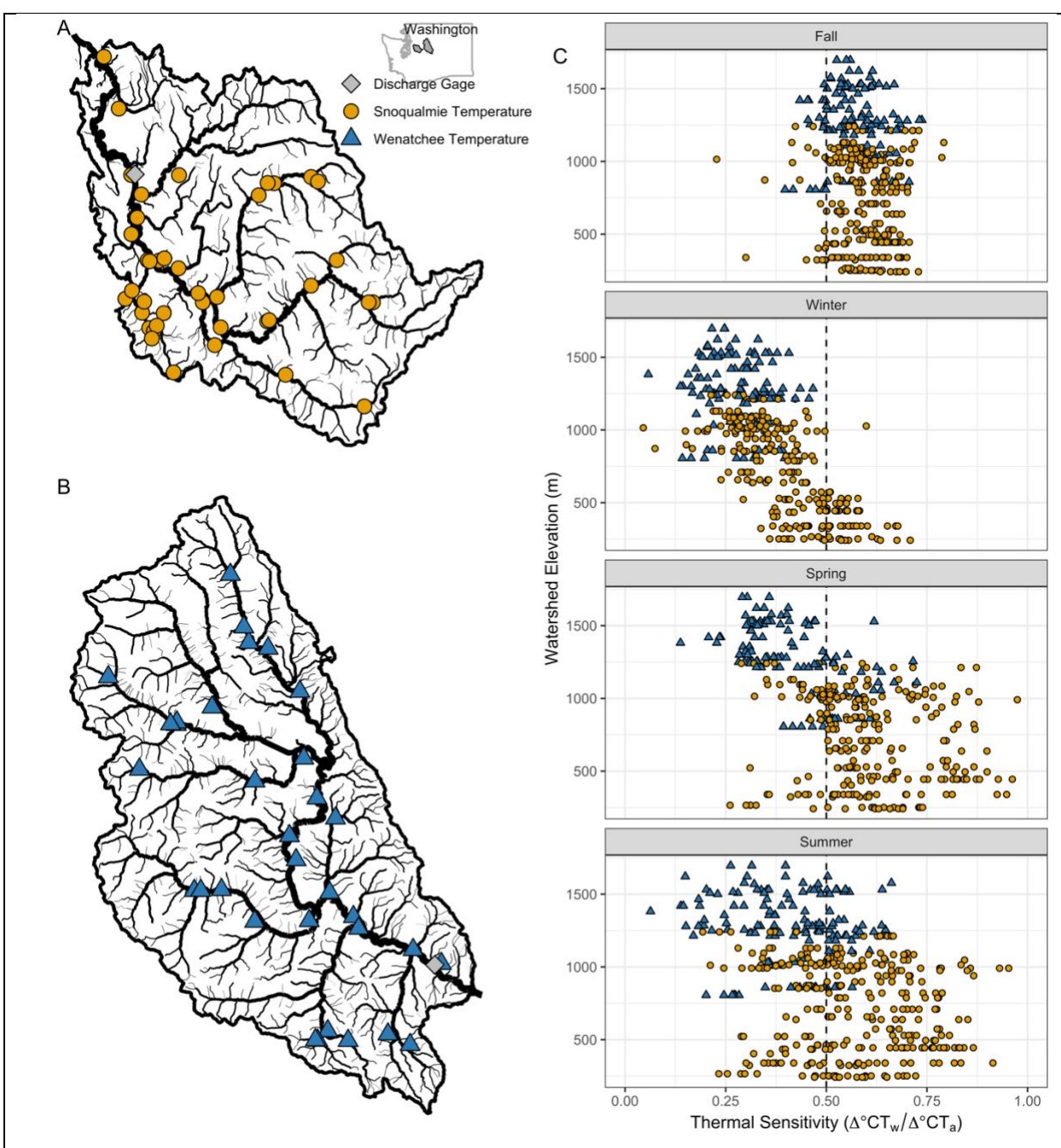

**Figure 1**. A map of the Snoqualmie (A) and Wenatchee (B) basins water and air temperature monitoring sites and the most downstream USGS gage for each basin. Thermal sensitivity, defined as the change in water temperature with a single degree change in air temperature, versus MWE for each site-year combination (C). The dashed line in Figure 1C is included as a reference.



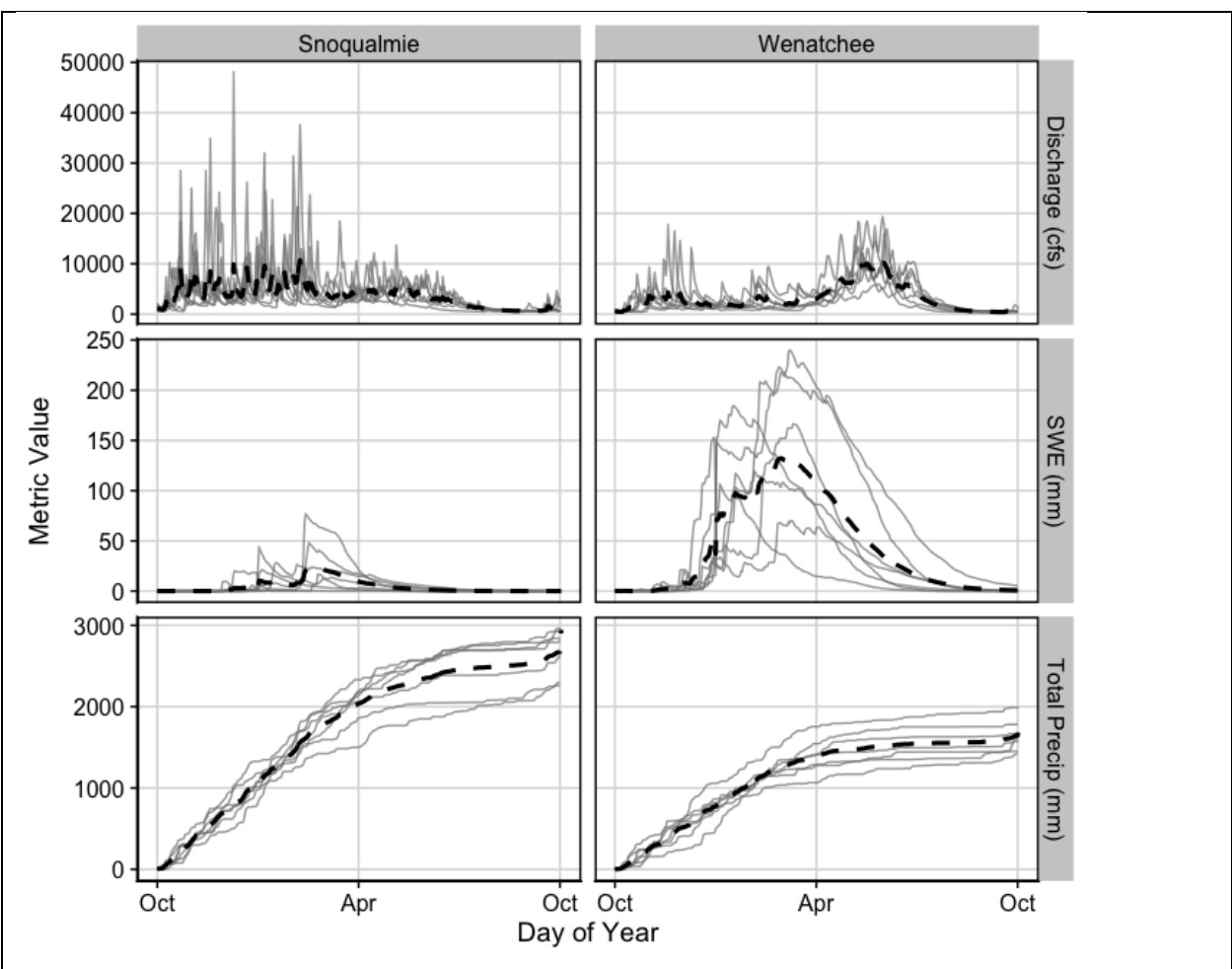

**Figure 2**. Average annual discharge, SWE, and total precipitation for the outlets of the Snoqualmie and Wenatchee basins across the sampling timeframe (black dashed lines) and interannual variability across the seven water years included in this analysis (gray lines). Discharge gage locations can be found in Figure 1A and 1B, and SWE and precipitation data is from DAYMET Daily Surface Weather data for the upstream watershed of each discharge gage (Thornton et al. 2020).


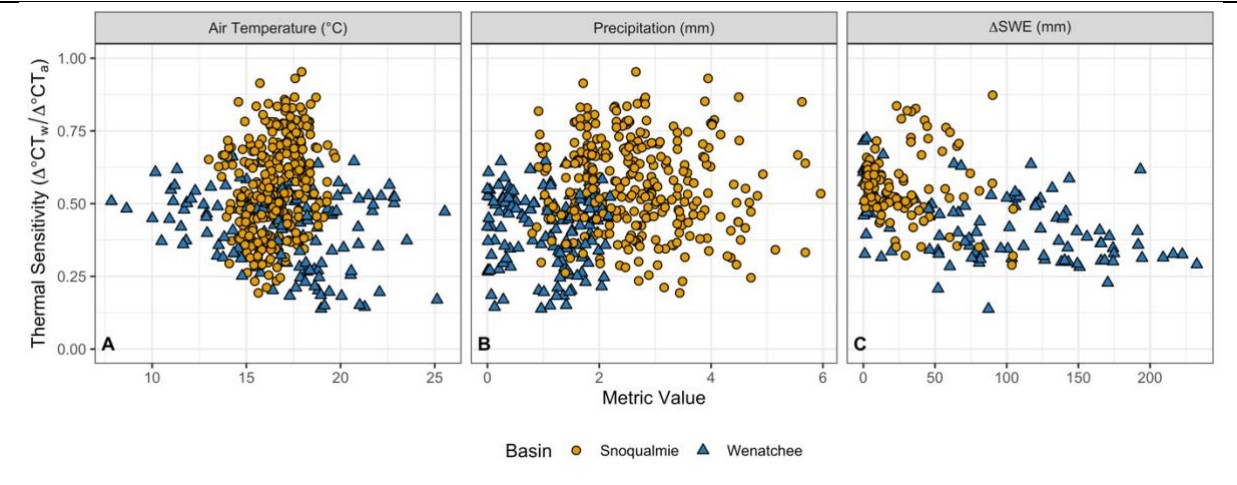

**Figure 3.** Summer thermal sensitivity values for all site-year combinations in the Snoqualmie and Wenatchee basins versus air temperature (A), and precipitation (B). Spring thermal sensitivity values for all site-year combinations versus total SWE (C) from gridded DAYMET data for each sampling point. Points are colored by basin. Basins that have no snowmelt in a given year are not shown on graph (C).


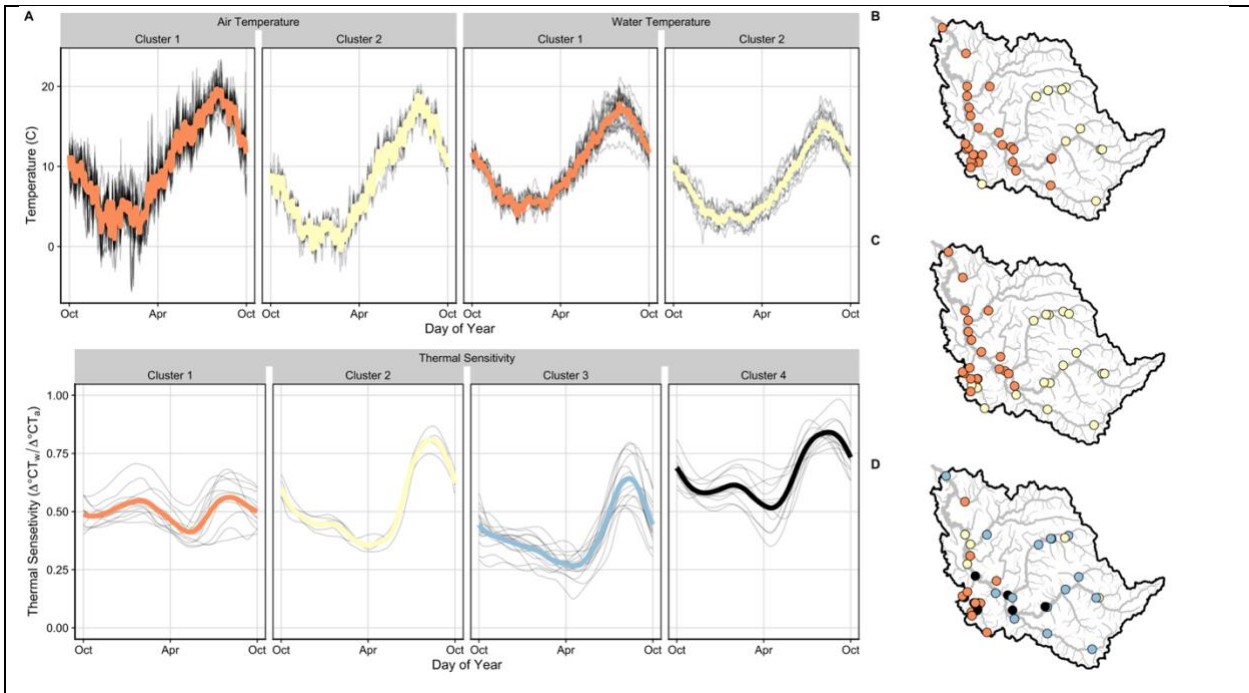

**Figure 4.** Average time series (A) and spatial clustering results (columns/colors indicate unique clusters) for average annual air temperature (B), water temperature (C), and thermal sensitivity (D) in the Snoqualmie basin. The spatial distribution for colored lines indicates mean average annual values for each cluster, and gray lines denote average annual values for each site within a given cluster.

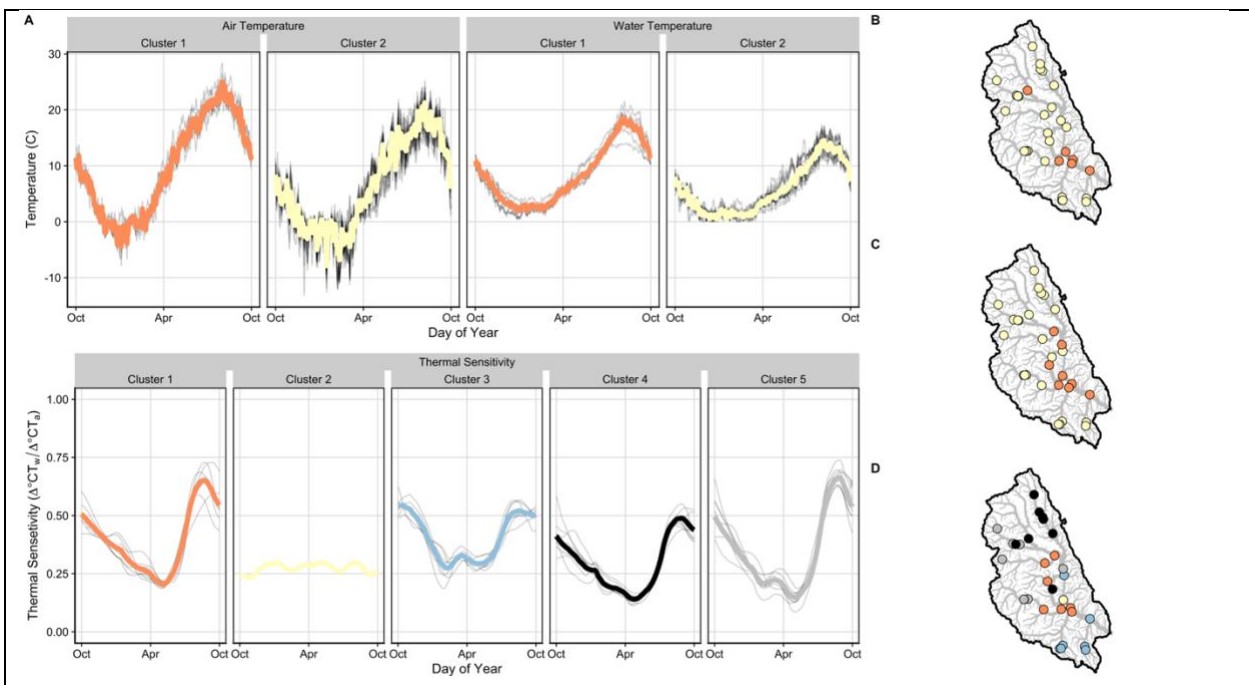

**Figure 5.** Average time series (A) and spatial clustering results (columns/colors indicate unique clusters) for average annual air temperature (B), water temperature (C), and thermal sensitivity (D) in the Wenatchee basin. The spatial distribution for colored lines indicates mean average annual values for each cluster, and gray lines denote average annual values for each site within a given cluster.

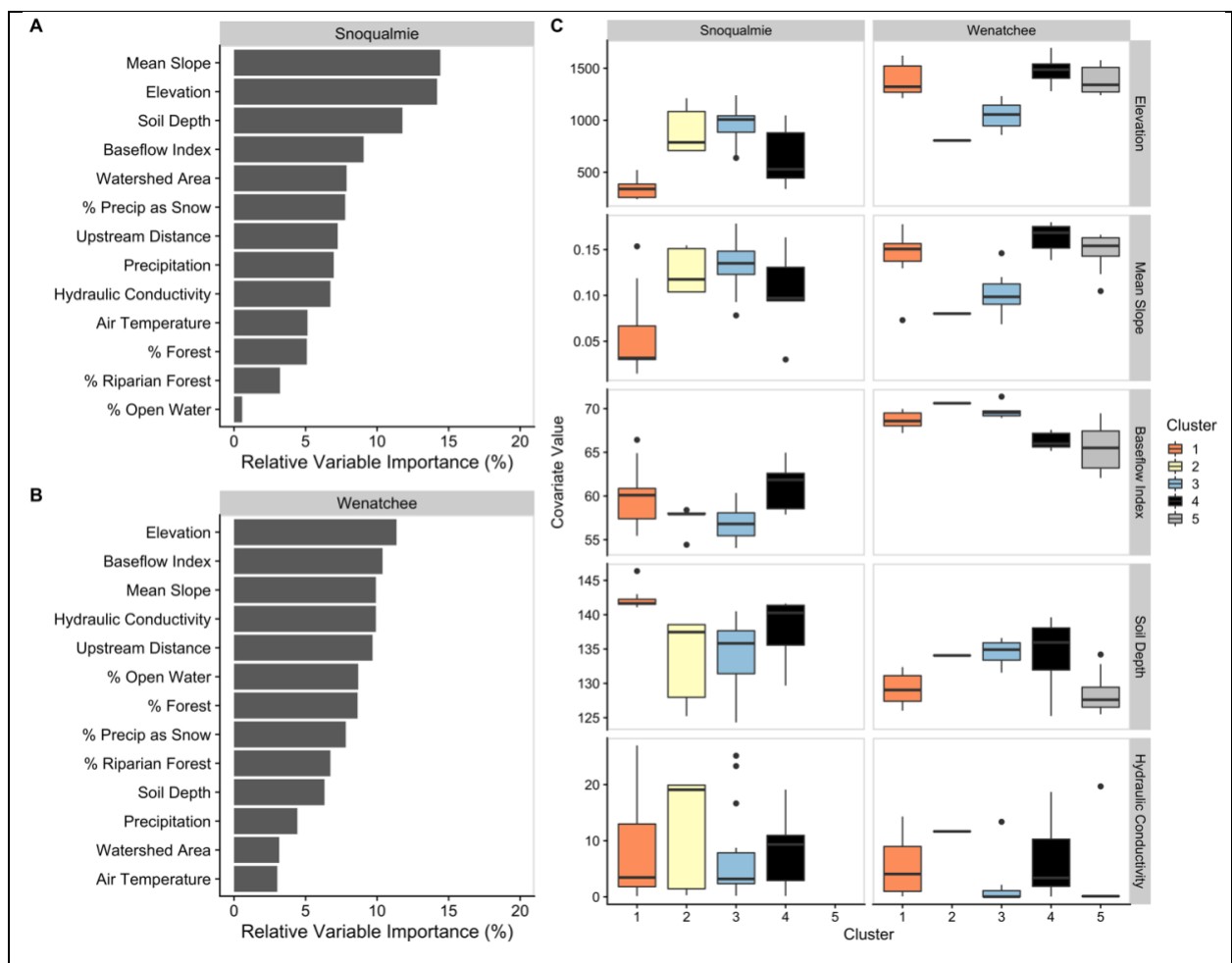

**Figure 6.** Relative variable importance for all covariates in the Snoqualmie (A) and Wenatchee (B) basins, and the distributions of variables across clusters for the four most important variables (C) in the Snoqualmie basin (Mean Slope, Elevation, Soil Depth, and Baseflow Index) and in the Wenatchee basin (Elevation, Baseflow Index, Mean Slope, and Hydraulic Conductivity). Boxes are grouped and colored by cluster membership. See Figure S8 for plots of the remaining relative variable importances.

**Table 1**. Hypothesized relationships between landscape covariates and thermal sensitivity based on previous literature (A) and the observed relationship between landscape variables and thermal sensitivities within our study basins in summer (B). Loess curves are shown to aid in visualization and correlation coefficients quantify the strength of the linear relationship. See Figure S6 for a detailed description of how river attributes covary with one another.

| A. Hypothesized Drivers | | | B. Observed Relationship |
|---|---|---|---|
| **Stream or watershed attribute (covarying variables)** | **Theoretical relationship with thermal sensitivity** | **Explanation** | **Observed Relationship in Summer** |
| Mean watershed slope<br><br>+elevation<br><br>+dist upstream<br><br>– soil depth | Negative | • Increased snowmelt and cooling due to faster velocity water movement and shorter water residence time (Winfree et al. 2018).<br>• Topographic shading associated with steep watersheds suppresses stream temperature by reducing exposure to solar radiation (Webb and Zhang 1997). |  |
| Mean watershed elevation<br><br>+slope<br><br>+dist upstream<br><br>+% lake area<br><br>– soil depth | Negative | • Higher elevations have higher snowmelt accumulation and greater proportion of snowmelt in spring.<br>• The impact of elevation on spring and early summer stream temperature is diminished in years with low winter snow accumulation. |  |
| Distance upstream<br><br>– watershed size<br><br>+slope<br><br>+elevation | Negative | • Duration of surface water's exposure to solar radiation and atmospheric energy flux is higher in low gradient watersheds with slower streamflow velocities (Poole and Berman 2001). |  |
| Percent riparian forest cover<br><br>+% forest cover<br><br>– watershed size | Negative | • Riparian vegetation provides shading to streams, reducing exposure to solar radiation (Webb and Zhang 1997), particularly during summer base flows.<br>• Forest canopy can influence snow accumulation within a watershed and snowmelt contribution to streams. Low density forests accumulate more snow relative to |  |

| | | | |
|---|---|---|---|
| | | high density forests (Varhola et al 2010).<br>• Conversion of forested land area can accelerate runoff and reduce infiltration, warming surface flows before they reach stream channels (Naiman et al. 2005; Nelson and Palmer 2007). | |
| Hydraulic Conductivity<br><br>+baseflow<br><br>index | Positive | • Hydraulic conductivity refers to the ability of a geologic material to transmit water and is calculated from mean lithological hydraulic conductivity content in surface or near surface geology.<br>• Relatively high hydraulic conductivity material would be represented by something like unconsolidated alluvial sands and gravels.<br>• High hydraulic conductivity is typically associated with areas of greater groundwater activity and lower, more stable thermal sensitivity values. |  |

**Table 2.** Physical environmental data and basin characteristics used to predict air-water clusters.

| Variable | Category | Units | Data Source |
|---|---|---|---|
| Watershed area | Basin Topography | km$^2$ | Hill et al. 2016 |
| Mean watershed elevation | Basin Topography | m | Hill et al. 2016 |
| Avg. stream slope | Basin Topography | mm$^{-1}$ | Hill et al. 2016 |
| Distance upstream | Basin Topography | km | Hill et al. 2016 |
| % Watershed forest | Land Use | % | Hill et al. 2016; Dewitz et al. 2019 |
| % Riparian forest | Land Use | % | Hill et al. 2016; Dewitz et al. 2019 |
| % Lake area | Land Use | % | Hill et al. 2016; Dewitz et al. 2019 |
| Avg. Temperature | Climate | C | Thornton et al. (2020) |
| Avg. Precipitation | Climate | mm | Thornton et al. (2020) |
| Avg. % precip as snow | Climate | % | Thornton et al. (2020) |
| Baseflow index | Hydrogeologic | % | Hill et al. 2016; Wolock 2003 |
| Hydraulic conductivity | Hydrogeologic | % | Hill et al. 2016; Olson and Hawkins 2014 |
| Soil depth to bedrock | Hydrogeologic | cm | Hill et al. 2016; Carlisle et al. 2009 |

15    **Table 3.** Air water correlation average summary metrics by basin and season. Averages are calculated as the mean value of summary metrics at all sites across each basin and season.

| | | Thermal Sensitivity | | | $R^2$ | | |
|---|---|---|---|---|---|---|---|
| | | Min | Mean | Max | Min | Mean | Max |
| Snoqualmie | Fall | 0.22 | 0.59 | 0.79 | 0.58 | 0.92 | 0.99 |
| | Winter | 0.05 | 0.40 | 0.71 | 0.20 | 0.86 | 0.96 |
| | Spring | 0.26 | 0.60 | 0.97 | 0.67 | 0.89 | 0.98 |
| | Summer | 0.19 | 0.56 | 0.95 | 0.41 | 0.85 | 0.97 |
| Wenatchee | Fall | 0.40 | 0.57 | 0.74 | 0.74 | 0.94 | 0.98 |
| | Winter | 0.05 | 0.28 | 0.47 | 0.44 | 0.84 | 0.95 |
| | Spring | 0.14 | 0.42 | 0.72 | 0.59 | 0.88 | 0.98 |
| | Summer | 0.06 | 0.41 | 0.66 | 0.08 | 0.77 | 0.96 |

**Table 4**. Averaged metrics for all sites within each cluster determined with the spatially weighted agglomerative hierarchical clustering. For timing metrics, days are reported as hydrologic day, where a value of 1 indicates October 1st and a value of 365 indicates September 30th.

| Metric | Basin | Cluster | # Sites | Mean | Minimum (timing) | Maximum (timing) | Cluster Stability |
|---|---|---|---|---|---|---|---|
| Thermal Sensitivity | Snoqualmie | 1 | 11 | 0.50 | 0.41 (224) | 0.56 (308) | 0.68 |
| | | 2 | 5 | 0.52 | 0.36 (181) | 0.81 (315) | 0.88 |
| | | 3 | 15 | 0.40 | 0.27 (201) | 0.64 (316) | 0.67 |
| | | 4 | 11 | 0.65 | 0.52 (199) | 0.84 (316) | 0.55 |
| | Wenatchee | 1 | 7 | 0.39 | 0.20 (216) | 0.65 (324) | 0.79 |
| | | 2 | 1 | 0.27 | 0.23 (28) | 0.30 (101) | 0.62 |
| | | 3 | 7 | 0.40 | 0.27 (131) | 0.54 (11) | 0.94 |
| | | 4 | 8 | 0.29 | 0.14 (207) | 0.48 (331) | 0.86 |
| | | 5 | 8 | 0.35 | 0.15 (214) | 0.66 (330) | 0.69 |
| Air | Snoqualmie | 1 | 31 | 10.2 | 1.01 (94) | 19.7 (305) | 0.91 |
| | | 2 | 11 | 8.02 | -0.42 (145) | 18.9 (304) | 0.73 |
| | Wenatchee | 1 | 6 | 9.68 | -4.52 (95) | 25.0 (304) | 0.95 |
| | | 2 | 25 | 6.48 | -7.88 (107) | 21.3 (310) | 0.85 |
| Water | Snoqualmie | 1 | 25 | 10.1 | 3.91 (94) | 17.8 (304) | 0.65 |
| | | 2 | 17 | 7.99 | 2.94 (94) | 15.6 (304) | 0.89 |
| | Wenatchee | 1 | 8 | 8.39 | 1.95 (108) | 18.5 (310) | 0.73 |
| | | 2 | 23 | 5.74 | 0.37 (107) | 14.5 (310) | 0.86 |

