# Peer review of "Empirical stream thermal sensitivities cluster on the landscape according to geology and climate"

_Hydrology and Earth System Sciences, 2022_

## Author Response (AR1)

Original reviewer comments are in blue.

Following HESS review policy, we initially replied to each of the reviewer's comments, but did not prepare a revised manuscript. These comments are in black.

After the HESS editor decision to revise and resubmit, we prepared a revised version of our manuscript. Specific changes to our manuscript are tracked in red.

**Reviewer 1 Comments**

In their study, McGill et al. characterized the thermal sensitivity of streams in two watersheds of the Pacific Northwest of the United States which describes how changes in stream temperature track changes in air temperature. They characterized thermal sensitivity using the conventional method looking at the slope between air and stream temperatures. They also used a novel approach using time-varying coefficients to capture how thermal sensitivity varies through the year – this is a truly interesting contribution to the field to assess in a continuous way the seasonality of thermal sensitivity. McGill et al. then performed a clustering analysis on the annual average time series of thermal sensitivities and used classification and regression trees to identify drivers of thermal sensitivity.

Overall, the manuscript is well written and beautifully illustrated. Methods are well described and sound. While results per se are mainly of regional interest, their use of time-varying coefficients offers a methodological contribution of interest to the journal. With a few revisions, this would make a quality contribution to HESS. The main points to address are the following:

We thank the reviewer for their interest and positive assessment of the topic and methodological approach. We also appreciate their thoughtful critiques, which we address below.

**1) Clearly present hypotheses/predictions underlying the study**
The manuscript identified three broad research questions (lines 82-85) and most of the results section then goes on to describe observed patterns found posteriori. Using a descriptive research approach is perfectly sound but I believe the study would be more informative if it used an explanatory research approach where the goal is to understand underlying causal mechanisms. In fact, at numerous places in the manuscript, authors talked of expected results: "we expected thermal sensitivity to increase with river size" (line 418), "we expected land cover characteristics such as open water and forest cover to be important predictors" (line 426) or "expectations of a negative relationship between thermal sensitivity and groundwater influence" (line 362). While not presented this way, it appears authors had hypotheses/predictions underlying their work.

I believe framing the manuscript to more clearly present hypotheses/predictions would make conclusions of broader interest to the stream temperature community in comparison to the current presentation of results which can be difficult to interpret without regional knowledge. For example, presenting results for the Chiwawa, White and Little Wenatchee rivers (lines 270), the tributaries to the mainstem and Raging River (line 259) or the Chumstick Creek (line 415) is factually correct but bears little meaning to someone unfamiliar with the study region.

There is a large body of work examining drivers of air and water temperature correlations, therefore we had numerous hypothesized drivers based on first principles and previous literature. The background work and these hypothesized drivers informed our decision about the suite of potential predictors to include. The drivers are often highly correlated, and we therefore attempted to summarize the structure of predicted drivers and their impacts on thermal sensitivity in Table 3. We chose to present the summary metric component as an exploratory analysis for a variety of reasons. First, exploratory research provides

a flexible framework for investigating complex and multifaceted topics, enabling the generation of novel ideas and hypotheses. Overreliance on hypothesis testing can pose dangers to the research process, including an overemphasis on statistical significance and p-hacking, which compromises the integrity and reproducibility of research findings (See Special Issue in The American Statistician 2019 Volume 73, Statistical Inference in the 21st Century: A World Beyond p < 0.05; Amrhein et al. 2019; Wasserstein & Lazar 2016). Importantly, the structure of our data lends itself more to an exploratory analysis than testing of a suite of individual hypotheses. Our study utilized a series of spatially distributed sites across the basin, and the configuration of these sites was designed to capture the range and variability of air and water temperature across the basin but not to test hypotheses about specific, causal mechanisms of thermal sensitivity. For example, ideally, if we wanted to test the impact of watershed slope on thermal sensitivity, we would have a series of more-or-less identical sites where only watershed slope varied between them to isolate slope as a driver. As variables across our basin are highly correlated, and our sample size only moderate, it would be difficult to parse apart the impact of specific drivers. We therefore believe that it is best not to frame our work in an explicit hypothesis testing framework for this manuscript.

However, as both reviewers brought up the same point, we clearly did not emphasize our statistical decision-making framework enough in our manuscript and will work to clarify it throughout. In particular, we will modify the methods paragraph on L127-133 to 1) explicitly state that our summary metric analysis was exploratory in nature to better understand patterns to set up future hypothesis testing, 2) ensure readers understand that relationships between thermal sensitivity and basin properties shown in Table 3 are hypotheses based on first principles that we lay out but do not explicitly test, 3) remove linear fits from Table 3 and instead include loess curves to aid the reader in visualization and avoid implying a regression was run, and 4) modify our phrasing of "summary metrics" results section accordingly.

Additionally, we agree that the discussion contains substantial regional knowledge that the average reader may not be familiar with. We believe that the details are important to include as they're often critical for understanding processes within a given basin and useful for local resource managers and practitioners. To provide context for readers, we will add a map with subbasin names, lakes, dams, and elevation as a supplementary figure. We will also modify lines where specific places are called out (e.g., Lines 259, 266, 270, etc.) so important elements about the basin are described, with the name of the location in parentheses to limit the necessity of regional knowledge to understand results.

We changed the title of section 2.2. to "Exploratory analysis of air-water correlation summary metrics" and modified L133-141 to state "A large body of literature examines landscape-level drivers of air and water temperature correlations within rivers. We therefore summarized hypothesized drivers of thermal sensitivity based on previous literature and their covarying landscape variables within our basins. We then conducted an exploratory analysis of the relationship between landscape covariates and thermal sensitivity to better understand patterns in our data and set up future hypothesis testing. Due to the correlated nature of our dataset, no formal statistical tests were conducted. We plotted summer thermal sensitivity metrics against hypothesized drivers, including mean watershed elevation (MWE), watershed slope, distance upstream, percent riparian forest cover, and substrate hydraulic conductivity. Loess curves were plotted to aid in data visualization, and correlation coefficients between thermal sensitivity and each landscape covariate were used to quantify the strength of the linear relationship. Covariate descriptions and sources are found in Table 1."

We also removed linear fits from the plots in Table 3 and now explicitly state that loess curves were included to aid in visualization in the caption to Table 3, and report correlation coefficients for each plot to describe the linear relationship between thermal sensitivity and landscape covariates. We amended L258-259 to explicitly state that "For landscape variables, correlation coefficients were overall small ($|\rho|$

< 0.3), indicating weak to non-existent linear relationships between landscape covariates and observed thermal sensitivity."

We have also included a supplementary figure with subbasin names, lakes, dams, and elevation to provide greater context for interested readers (Figure S4 and S5).

Along the same lines, authors performed three distinct cluster analyses (air temperature, water temperature, thermal sensitivity) and their goal was unclear until I reached the discussion and understood that they wished to show that thermal sensitivity clusters offers additional information to what we find when studying solely air/stream temperatures. If that was one of the goals of the study, I suggest it be clearly stated and communicated as a take-home message.

We will amend the manuscript to include this. We will modify L83-84 from "What are the characteristic regimes of air-water temperature correlations and how do they cluster on the landscape?" to "What are the characteristic regimes of air-water temperature correlations, how do they cluster on the landscape, and how do they differ from clusters of only air and water temperature?" to highlight that this was a goal of our study. Furthermore, we will add a sentence in L181 (Agglomerative Hierarchical Clustering in the Methods section) reiterating that we ran three distinct clustering analyses.

We modified L87 accordingly and reiterated our intention to compare clustering of thermal sensitivity, air, and water temperature in Section 2.3, L150-153 and L198-200.

**2) Conclusions need to be better supported by analyses and results.**
A few statements in the results are not sufficiently supported by analyses. Moreover, the results section often lacks precision and statements are often little quantified. For example, the abstract states that thermal sensitivity regimes "differed in both timing and magnitude of sensitivity" and while Figures 4-5 offer a nice illustration of regimes, no formal analysis clearly compared the timing/magnitude of clusters. There are a few other examples in the results section:

We will modify Table 4, which currently includes measures of the cluster-specific thermal sensitivity range, mean, and stability, to also include timing for maximum and minimum thermal sensitivity.

Table 4 formerly reported the mean, maximum, and minimum thermal sensitivity for all individual stations within a cluster. We modified this slightly to report the cluster-average mean, maximum, and minimum to improve clarity and facilitate comparison across different clusters more easily. We also included the timing of the cluster-average minimum and maximum values and cluster-averages for air and water clusters in addition to thermal sensitivity.

We included several quantitative metrics from Table 4 to describe cluster differences in Section 3.2, L269-301. A few examples from this paragraph are described below.
      1) L275-276 we state, "For example, within both basins seasonal water temperatures were synchronized, with the cluster minimum and maximum water temperatures occurring within a day of each other (Table 4)."
      2) L288-289 we state, "Cluster 2 was characterized by a mean thermal sensitivity of 0.52 and the highest annual variability, with a cluster-average range of 0.45".
      3) L294-296 we state, "Clusters 1, 4, and 5 demonstrated similar seasonal patterns in thermal sensitivities, with minimum values occurring in late Spring (water days 216, 207, 214) and maximum values occurring in late summer (water days 324, 331, 330)."

The manuscript states that "thermal sensitivity estimates were not entirely consistent" (line 230) although it is not clear what consistent refers to and if it was quantified. Similar wording regarding a "consistent seasonal signal" (line 243) should be revised.

We will modify our use of the word consistent to identical.

We have made the requested change. Furthermore, the figure this sentence describes is now in a separate appendix (Appendix A) devoted to a description of interannual variability within our dataset.

The manuscript states that "only SWE displayed a relationship with thermal sensitivity" (line 233) while no formal analysis was done – a visual assessment is not sufficient to determine the presence/absence of a relationship. At minimum, R² values should be presented in Figure 3 to assess the strength of relationships. I also question here the presence of the "wedge-shaped pattern" (line 234) for SWE in Figure 3 which does not stand out clearly and may be simply due to fewer data points for large SWE values. Further analysis is required to assess the strength of this relationship.

Our "thermal sensitivity metrics" section was exploratory in nature due to the structure of our data, which we will more clearly reiterate in the manuscript (see the above response for further details).

See above for a more detailed description of completed changes. The phrase was edited to state "only SWE appeared to have a linear relationship with thermal sensitivity."

Similarly, I question some of the relationships between variables that are discussed in Table 3. It is not clear to me that we can see a "consistent negative relationship between thermal sensitivity, distance upstream and MWE" (line 236). Figures in Table 3 do not present the correlation coefficient but my visual assessment is that it is likely close to 0 for MWE. The results section also points towards a "weakly positive and parabolic" (line 238) relationship between hydraulic conductivity and thermal sensitivity in the Snoqualmie basin, yet a linear regression is plotted in the figure in Table 3. Overall, many different relationships between thermal sensitivity and environmental variables appear to be weak and should be confirmed using statistical analyses.

We thank the reviewer for pointing out that including linear fits to the data in Table 3 suggests that regressions were run; we will remove the lines from this table and instead add loess curves, in addition to explicitly stating that no regressions were run and our summary metric was exploratory in nature. Given that we don't necessarily expect linear relationships, we are hesitant to include correlation coefficients, particularly when adding a nonlinear smoothed line to aid visualization.

Furthermore, we agree that patterns are often weak and inconsistent, and explicitly state "Overall, weak and inconsistent patterns emerge in summer between thermal sensitivity and landscape and climate variables (Figure 3; Table 3)" on Lines 232-234. Inherent covariation in river basins can hinder statistical efforts to identify mechanistic links between landscape gradients and features of aquatic ecosystems (Lucero et al. 2011); variables may have a small impact that went undetected due to noisy observations or limited variability within our study region (Lines 428-440 discuss this in the manuscript). We nevertheless thought it important to include the summary metric analysis in our results, as these covariates are assumed to be important controls on thermal sensitivity, and we aimed to clearly set up a framework in which future studies could conduct more targeted analyses.

We removed linear fits from the plots in Table 3 and now explicitly state that loess curves were included to aid in visualization in the caption to Table 3, and report correlation coefficients for each plot to describe the linear relationship between thermal sensitivity and landscape covariates. We amended L258-259 to explicitly state that "For landscape variables, correlation coefficients were overall small ($|\rho| < 0.3$),

indicating weak to non-existent linear relationships between landscape covariates and observed thermal sensitivity.”

Last, the paragraph from lines 256-275 should be more precise and quantify some statements such as “somewhat high mean thermal sensitivities” (line 263), “overall high thermal sensitivity and low variability” (line 267), “cluster 3 had the greatest variability through time” (line 271)

We will add more specific text to this section, in particular referencing a modified Table 4 to include exact values for cluster-specific thermal sensitivity range, mean, stability, and timing for maximum and minimum thermal sensitivity.

We have modified this section to include more specific descriptions of cluster properties, particularly drawing from cluster-average values listed in Table 4. A few examples from this paragraph are described below, but please see the revised manuscript to view all changes.
    1) L275-276 we state, “For example, within both basins seasonal water temperatures were synchronized, with the cluster minimum and maximum water temperatures occurring within a day of each other (Table 4).”
    2) L288-289 we state, “Cluster 2 was characterized by a mean thermal sensitivity of 0.52 and the highest annual variability, with a cluster-average range of 0.45”.
    3) L294-296 we state, “Clusters 1, 4, and 5 demonstrated similar seasonal patterns in thermal sensitivities, with minimum values occurring in late Spring (water days 216, 207, 214) and maximum values occurring in late summer (water days 324, 331, 330).”

**3) Better consideration of interannual variability in thermal sensitivity**
The cluster analysis of thermal sensitivity relies on an annual average time series of thermal sensitivities. I suggest that the manuscript better lay out the implications of having sites with fewer years of data. Did this have a strong influence on the clustering? For example, was the clustering similar if performed using a single and most common year of data available? Section 4.5 in the discussion does a very good job of discussing limitations in general terms but adding a more formal analysis would be more convincing.

The reviewer brings up a good point that we don't consider interannual variability explicitly in our clustering analysis. Ideally, we would be able to run our clustering algorithm for each year individually to assess how clusters differ across specific years. However, the issue that arises is that the set of sites with continuous data can be quite different between years, limiting our potential to compare across years. Therefore, to assess cluster sensitivity to interannual variability, we will use a “leave-one-out” approach similar to the stability analysis outlined in Lines 199-205 (assessing cluster stability when leaving out sites). We will leave one year out when calculating average annual time series, and subsequently run the clustering analysis on the annual average time series for N-1 years of data. We will then compare cluster similarity to our results reported for all years of data. This analysis will be completed for each year we have data for. Results from the analysis will allow us to assess if any specific years have a particularly strong influence on clustering results (i.e., clustering results differ substantially when data from 20XX is removed). Results will be reported in detail in the supplementary material, and implications of the sensitivity analysis will be discussed in the “caveats and limitations section” of the discussion. This assessment will be a first step towards more comprehensively assessing interannual variability. Results can also be compared to a large body of work assessing interannual variability of water temperature, particularly in the Snoqualmie River (Steel et al. 2019).

We have completed a sensitivity analysis to assess if the removal of data from a specific water year had a particularly strong influence on clustering results. Results indicated that, although cluster agreement between our reported results and the reduced dataset was not perfect, our analysis using average annual

data generally captured broad patterns within our dataset. More detail on this new analysis is included in the revised Appendix A of our manuscript.

**MINOR POINTS**

line 52: Define thermal memory as it is not a widely accepted concept.

We will make the requested change.

We have changed "thermal memory" to "annual hysteresis" as we believe this is a more widely accepted term.

line 179: what is the dissimilarity matrix d^c xy ?

This is notation used to reference the spatially weighted dissimilarity matrix, whereas dxy references the Canberra distance matrix. We will clarify this in the methods.

We have added the phrase "$d_{xy}^c$ is the spatially weighted dissimilarity matrix" to L182 in our manuscript.

line 246: Although presented in Supplementary Material (Table S2), I suggest adding a sentence to give an idea of the variability in the number of clusters according to the method used.

We will make the requested change.

We have added the range of clusters suggested by cluster validity indices to L269-272.

line 249: Without regional knowledge, it is not clear from figures that "air and water temperature correspond closely with elevational gradients".

We will add elevation shading to the basins in a new supplementary figure map to clarify this point.

We have added supplementary figures S4 and S5 showing elevation gradients for each basin.

line 302: thermal sensitivities varied substantially between sites? I suggest being more explicit as to what is being compared here.

We will modify this sentence to "Fall thermal sensitivities were relatively homogeneous, with 90% of values falling between 0.47 and 0.70, whereas spring and summer thermal sensitivities exhibited a broader range of values, with 90% of values falling between 0.30 and 0.84 in spring and 0.25 and 0.78 in summer."

We modified L248-253 in the results to "Fall thermal sensitivities were relatively homogeneous, with 90% of values falling between 0.47 and 0.70, whereas spring and summer thermal sensitivities exhibited a broader range of values, with 90% of values falling between 0.30 and 0.84 in spring and 0.25 and 0.78 in summer" to better support this statement in the discussion.

line 306: non-redundant aspects relative to what? I suggest being more explicit as to what is being compared here.

We will modify this sentence to "Thermal sensitivity regimes reflect non-redundant aspects of river dynamics relative to air and water temperature alone."

We modified this sentence to "Thermal sensitivity regimes reflect non-redundant aspects of river dynamics relative to air and water temperature alone."

line 323: This statement is a bit strong and little supported by results. For example, static thermal sensitivity (e.g. Table 2) may in fact align well with clusters defined using the time-varying approach, something the manuscript did not look into.

The way thermal sensitivity is typically measured, it is often conceptualized as a single, stationary value, rather than an average of multiple estimates. We believe that this is an important distinction; recognizing that a parameter shifts over time and using the average is fundamentally different from assuming a parameter is static through time. Our point here was that recognizing variability in this parameter is important, and we will work to clarify this in the manuscript.

We have modified L313-321 to state "Thermal sensitivity varies throughout the year and reflects hydrologic conditions at a given time and place within a watershed; therefore, it should not be conceptualized as a static value. Although summary metrics of thermal sensitivity, such as average values over the summer, can still prove useful and informative, it is essential to acknowledge the non-stationarity of the relationship between air and water temperature to obtain a more accurate understanding of how river temperature responds to changing conditions."

line 334: To what does the buffering refer to?

Buffering refers to the process wherein snowmelt-influenced streams have lower thermal sensitivity (i.e., buffering against climate variability). This is due to a direct input of cold water and a corresponding increase in flow rates and depths which mitigates the impact of surface heat exchanges by increasing thermal inertia (van Vliet et al. 2011; Siegel et al. 2022). We will work to clarify this in the manuscript.

We have modified L366-368 to state "Importantly, snowmelt buffering, the process wherein snowmelt-influenced streams have lower thermal sensitivity due to a direct input of cold water and a corresponding increase in flow rates and water depths (van Vliet et al. 2011, Siegel et al. 2022), diminishes throughout the summer."

line 335: A comma is missing after "summer"

We will make the requested change.

We completed the requested change.

line 361: Do "summary metric regression" refer to Table 2?

Yes, the summary metrics refer to Table 2, however, this is a mistake in wording on our part. We will amend the sentence to state "… results from the summary metric exploratory analysis were mixed…".

We completed the stated change.

line 435: Are there large dams in the two studied basins? If so, it should be clearly stated as this could explain why certain environmental variables had little influence.

There is a dam and reservoir on a major tributary to the Snoqualmie River, the Tolt River. Several small dams exist on tributaries to the Wenatchee River, and a large lake (Lake Wenatchee) sits at the junction of

the White and Chiwawa Rivers. We will include all basin names, lakes, and dams on a map in the supplementary material and reference their potential to influence results in the manuscript.

We have included supplementary figures (Figure S4 and S5) with the location of the single reservoir and large lake within our basin.

line 457: What were the bandwidth and averaging periods used? I couldn't find this information anywhere in the methodology.

We thank the reviewer for pointing out this omission. We will include the bandwidth used in the methods section.

We have included the bandwidth utilized (0..2) in the methods on L164.

Citations

Amrhein, V., Greenland, S., McShane, B. 2019. Scientists rise up against statistical significance. Nature, 567: 305-307, DOI: https://doi.org/10.1038/d41586-019-00857-9.

Wasserstein, R.L. & Lazar, N.A. 2016. The ASA statement on p-values: context, processes, and purpose. The American Statistician, 70(2): 129-133, DOI: https://doi.org/10.1080/00031305.2016.1154108.

Steel, E.A., Marsha, A., Fullerton, A.H., Olden, J.D., Larkin, N.K., Lee, S.Y., Ferguson, A. 2018. Thermal landscapes in a changing climate: biological implications of water temperature patterns in an extreme year. Canadian Journal of Fisheries and Aquatic Sciences, 76(10): 1740-1756, DOI: https://doi.org/10.1139/cjfas-2018-0244.

Lucero, Y., Steel, E.A., Burnett, K.M., Christiansen, K. 2011. Untangling Human Development and Natural Gradients: Implications of Underlying Correlation Structure for Linking Landscapes and Riverine Ecosystems. River Systems, 19(3): 207–24, DOI: https://doi.org/10.1127/1868-5749/2011/019-0024.

van Vliet, M.T.H., Ludwig, F., Zwolsman, J.J.G., Weedon, G.P., Kabat, P. 2011. Global river temperatures and sensitivity to atmospheric warming and changes in river flow. Water Resources Research, 47:W02544, DOI:  https://doi.org/10.1029/2010WR009198.

Siegel, J.E., Fullerton, A.H., Jordan, C.E. 2022. Accounting for snowpack and time-varying lags in statistical models of stream temperature. Journal of Hydrology X, 17: 100136, DOI: https://doi.org/10.1016/j.hydroa.2022.100136

Overall, I really like this study, from the conceptual development, to the data collection, to much of the analysis (especially continuous time series of stream thermal sensitivity), and discussion. I think there is great transferrable value of interest to HESS readership. I have some criticisms of the way the sensitivity metric data are visualized and discussed in Figs 1 and 3, but I really like the metric time series analysis is shown in Fig 4 and 5.

We thank the reviewer for their positive assessment of our manuscript. We also appreciate their thoughtful critiques, which we address below.

It would be nice to show representative streamflow from those basins over the same time periods to help assess how thermal sensitivity may be driven by the volume of water in the channel at any one time (determines channel water thermal inertia to changes in net heat flux). Low stream discharge volume may be a primary driver of increased thermal sensitivity at many sites in late summer, though I do not see discharge included in any of your quantitative analysis of controlling parameters (though baseflow index is derived from stream discharge, and is included here in a general way).

We agree that streamflow likely impacts thermal sensitivity, particularly in the dry summer months when discharge is lowest, temperatures highest, and features such as groundwater seeps may show up clearly. Discharge was not included in our analysis due to the lack of spatially and temporally resolved streamflow data across the basins. There are relatively few USGS and locally maintained discharge gauges in the Snoqualmie and Wenatchee basins, and most gauges do not directly correspond to our temperature sites. Watershed area is likely the best proxy for average annual discharge, with baseflow index loosely corresponding to specific discharge in summer. We agree that representative time series of discharge would be useful for readers and will include average discharge at the outlet of each basin as a panel on Figure 1. The location of these outlet gauges is already shown on the maps.

We have included annual time series of discharge, SWE, and precipitation for the outlet of the Snoqualmie and Wenatchee basins in a new figure, Figure 2.

As mentioned by Reviewer 1, given the 'expectations' listed in Table 3 it would be nice to frame the study as hypothesis driven/testing, which would not be a major change to what you have now. Below I list some more major and minor points that could be considered during the revision process.

There is a large body of work examining drivers of air and water temperature correlations, therefore we had numerous hypothesized drivers based on first principles and previous literature. The background work and these hypothesized drivers informed our decision about the suite of potential predictors to include. The drivers are often highly correlated, and we therefore attempted to summarize the structure of predicted drivers and their impacts on thermal sensitivity in Table 3. We chose to present the summary metric component as an exploratory analysis for a variety of reasons. First, exploratory research provides a flexible framework for investigating complex and multifaceted topics, enabling the generation of novel ideas and hypotheses. Overreliance on hypothesis testing can pose dangers to the research process, including an overemphasis on statistical significance and p-hacking, which compromises the integrity and reproducibility of research findings (See Special Issue in The American Statistician 2019 Volume 73, Statistical Inference in the 21st Century: A World Beyond $p < 0.05$; Amrhein et al. 2019; Wasserstein & Lazar 2016). Importantly, the structure of our data lends itself more to an exploratory analysis than testing of a suite of individual hypotheses. Our study utilized a series of spatially distributed sites across the basin, and the configuration of these sites was designed to capture the range and variability of air and water temperature across the basin but not to test hypotheses about specific, causal mechanisms of thermal sensitivity. For example, ideally, if we wanted to test the impact of watershed slope on thermal

sensitivity we would have a series of more-or-less identical sites where only watershed slope varied between them to isolate slope as a driver. As variables across our basin are highly correlated, and our sample size only moderate, it would be difficult to parse apart the impact of specific drivers. We therefore believe that it is best not to frame our work in an explicit hypothesis testing framework for this manuscript.

However, as both reviewers brought up the same point, we clearly did not emphasize our statistical decision-making framework enough in our manuscript and will work to clarify it throughout. In particular, we will modify the methods paragraph on L127-133 to 1) explicitly state that our summary metric analysis was exploratory in nature to better understand patterns to set up future hypothesis testing, 2) ensure readers understand that relationships between thermal sensitivity and basin properties shown in Table 3 are hypotheses based on first principles that we lay out but do not explicitly test, 3) remove linear fits from Table 3 and instead include loess curves to aid the reader in visualization and avoid implying a regression was run, and 4) modify our phrasing of "summary metrics" results section accordingly.

We changed the title of section 2.2. to "Exploratory analysis of air-water correlation summary metrics" and modified L133-141 to state "A large body of literature examines landscape-level drivers of air and water temperature correlations within rivers. We therefore summarized hypothesized drivers of thermal sensitivity based on previous literature and their covarying landscape variables within our basins. We then conducted an exploratory analysis of the relationship between landscape covariates and thermal sensitivity to better understand patterns in our data and set up future hypothesis testing. Due to the correlated nature of our dataset, no formal statistical tests were conducted. We plotted summer thermal sensitivity metrics against hypothesized drivers, including mean watershed elevation (MWE), watershed slope, distance upstream, percent riparian forest cover, and substrate hydraulic conductivity. Loess curves were plotted to aid in data visualization, and correlation coefficients between thermal sensitivity and each landscape covariate were used to quantify the strength of the linear relationship. Covariate descriptions and sources are found in Table 1."

We also removed linear fits from the plots in Table 3 and now explicitly state that loess curves were included to aid in visualization in the caption to Table 3, and report correlation coefficients for each plot to describe the linear relationship between thermal sensitivity and landscape covariates. We amended L258-259 to explicitly state that "For landscape variables, correlation coefficients were overall small ($|\rho| < 0.3$), indicating weak to non-existent linear relationships between landscape covariates and observed thermal sensitivity."

1. L15: '…it is critical to both understand the underlying processes causing stream warming and identify the streams most and least sensitive to environmental change.' Measurement of air-water temperature relations across the landscape provides an efficient way to address this important topic. However, it is a localized measurement that may not reflect general behavior across the stream system as other related studies have shown, especially when there is strong variability in groundwater discharge (eg Z. Johnson et al papers). This point is discussed somewhat in the body text, but still could be made more clear throughout. Local stream channel heat exchange process can dominate the local air-water temp sensitivity metrics, which speaks to collecting spatially distributed datasets, as you nicely did for this study.

We agree with the Reviewer's point that air-water temperature measurements can be localized in space and time, and believe our manuscript highlights this fact throughout. We will emphasize the fact that local stream channel heat exchange processes such as groundwater inflow can be a dominant control on thermal sensitivity in certain situations.

2. Although stream thermal sensitivity is quantified relative to changes in air temperature, air temperature warming may not always be the primary driver of stream temperature warming. Sensible heat fluxes are often dwarfed by solar and latent heat fluxes along the stream corridor. L39 acknowledges this important point. However, climate warming as typically described is primarily driven by the impacts on the global long wave radiation budget by accumulation of greenhouse gasses, not changes in solar short wave radiation input. The point that air temperature itself may not be the primary driver of stream temperature change at the seasonal timescale should be more clear, throughout. For example there is this statement on L122: 'The slope of this relationship, the thermal sensitivity, indicates how sensitive a given stream's water temperature is to changes in air temperature.' I am not sure that is true, more that air and stream temperature are sensitive to solar radiation in more or less coupled ways. This is kind of a nuanced point, but I have interacted with several people who interpret these type of metrics as air temperature often being the primary driver of stream temperature, presumably through sensible heat exchange.

The reviewer brings up an excellent point that air and water temperatures are correlated primarily due to a similar response to solar radiation, not because air temperature drives water temperature. This is a point we want to emphasize to readers, and we will amend L122 to more accurately reflect this and attempt to make it clear throughout the manuscript. We thank the reviewer for the suggested wording.

We have modified L125-128 to state "The slope of this relationship, the thermal sensitivity, indicates the average difference in water temperature when comparing time periods with a one-degree difference in air temperature. For example, a thermal sensitivity of 0.5 would indicate that, based on historical data, when air temperature at a site differs by 1°C, water temperature differs on average by 0.5°C (Leach and Moore 2019)". This new phrasing avoids implying that air temperature controls water temperature.

3. L41 and elsewhere: Addition of water to the stream channel impacts thermal inertia and stream temperature sensitivity, even if that water is of the same temperature as the channel. How are these patterns impacted by variable stream discharge at locations over time and along the stream network continuum? For example, clusters 2,3, and 4 show substantial increases in thermal sensitivity in late summer during presumably the lowest flows.

We agree that high thermal sensitivity in summer is likely mediated by low discharge, as in both the Snoqualmie and Wenatchee basins discharge is lowest in late summer. We will emphasize this in the manuscript by adding discharge time series at the outflow of each basin to Figure 1 and stating that low summer discharge values likely contribute to increased thermal sensitives in late summer in L328-341 of the discussion.

Figure 2 was added to the manuscript to illustrate basin-wide discharge regimes.

We have modified L505-508 to state "For many of our study sites, thermal sensitives were highest in late summer during the hottest, lowest flow portion of the year. Previous studies have found that the impact of fluctuations in discharge generally increases during dry, warm periods, when rivers have a lower thermal capacity and are more sensitive to atmospheric warming (van Vliet et al. 2013)."

4. I found the 'Identification of environmental drivers in thermal sensitivity' section most questionable given the relatively small sample size and lack of representation across varied types of watersheds. Also, hydrologic attributes downstream in a network are inherently influenced by physical attributes upgradient in the network, and your spatial sampling spans upstream to downstream. I think that statements such as: 'Annual patterns in thermal sensitivity are largely controlled by underlying geology and climate across two Pacific Northwest river basins' are too definitive given the sparse nature of the datasets across a range of geologic and climatic variables.

It may be that stream network position is more important that some of the apparent shifts in the tested physical variables.

We will amend this sentence to say "Underlying geology and climate are important controls on annual patterns in thermal sensitivity across two Pacific Northwest river basins", which more accurately reflects the results of our CART analysis. We include both upstream distance and watershed area in our examined covariates for the clustering analysis, both of which had middling-to-low importance.

We amended L317-319 to state "Underlying geology and climate are important controls on annual patterns in thermal sensitivity across two Pacific Northwest river basins." Additionally, we intentionally limited our conclusions to the Snoqualmie and Wenatchee basins, as we do not feel that we sufficiently sampled across a broad enough range of geology and climate variables to draw general conclusions.

5. The air-water temp sensitivity metrics in Fig 1 are somewhat difficult to interpret, as data are plotted seasonally over years for individual sites all by elevation. Given some sites appear at quite similar elevation, its not possible to disentangle changes by site and changes by elevation, and which sites are upstream/downstream of each other. I do not have any great advice with how to deal with this, however. Different colors for all sites would be overwhelming. Apparent trends in thermal sensitivity with elevation in some seasons may be somewhat of an artifact of plotting both watershed datasets together. Taken alone, seasonal datasets from either watershed would not seem to show an increasing trend with elevation. Given the inherent hydrogeological and climate differences between the two study watersheds I am not sure it is appropriate to depict and analysis the season metrics together.

We acknowledge that it can be difficult to show all aspects of the data in a single plot; it was not our intent to show interannual differences or upstream-downstream effects with this figure, but rather to visualize general patterns within and across river basins. Comparing across basins can be a powerful tool and is a common practice in hydrologic sciences, and our inclusion of differing colors for the basins was designed to acknowledge that basic-specific differences exist beyond the parameter (elevation) shown.

6. There are numerous places in the paper where a statistical test is inferred but it is not clear if a statistical test (along with p-value) was performed. For example: L233 'Overall, weak and inconsistent patterns emerge in summer between thermal sensitivity and landscape and climate variables'. While 'patterns' does not indicate a test, 'weak' does. Also, L230 'Thermal sensitivities for sites with consistent data coverage tended to covary,..'. Covariance is a statistical test and should be associated with a significance level. My biggest problem is with the fourth column of Table 4, where linear fits are shown to the datasets without significance levels being directly indicated. I am pretty sure that many of those fits are not significant, and therefore should certainly not be shown. Plotting the best fit lines tends to influence the reader's perception of trends, and if they are not statistically significant, they do now exist according to those significance metrics (eg p value levels). Labeling the column 'observed relationship' indicates all linear fits shown are significant and I see that as highly problematic.

See the above comment for a more detailed response to the themes addressed in this comment. In short, we will modify the methods paragraph on L127-133 to 1) explicitly state that our summary metric analysis was exploratory in nature to better understand patterns to set up future hypothesis testing and that no statistical tests were performed, 2) ensure readers understand that relationships between thermal sensitivity and basin properties shown in Table 3 are hypotheses based on first principles that we lay out but do not explicitly test, 3) remove linear fits from Table 3 and instead include loess curves to avoid implying a regression was run, and 4) modify our phrasing of "summary metrics" results section accordingly.

We changed the title of section 2.2. to "Exploratory analysis of air-water correlation summary metrics" and modified L133-141 to state "A large body of literature examines landscape-level drivers of air and water temperature correlations within rivers. We therefore summarized hypothesized drivers of thermal sensitivity based on previous literature and their covarying landscape variables within our basins. We then conducted an exploratory analysis of the relationship between landscape covariates and thermal sensitivity to better understand patterns in our data and set up future hypothesis testing. Due to the correlated nature of our dataset, no formal statistical tests were conducted. We plotted summer thermal sensitivity metrics against hypothesized drivers, including mean watershed elevation (MWE), watershed slope, distance upstream, percent riparian forest cover, and substrate hydraulic conductivity. Loess curves were plotted to aid in data visualization, and correlation coefficients between thermal sensitivity and each landscape covariate were used to quantify the strength of the linear relationship. Covariate descriptions and sources are found in Table 1."

We also removed linear fits from the plots in Table 3 and now explicitly state that loess curves were included to aid in visualization in the caption to Table 3, and report correlation coefficients for each plot to describe the linear relationship between thermal sensitivity and landscape covariates. We amended L258-259 to explicitly state that "For landscape variables, correlation coefficients were overall small ($|\rho| < 0.3$), indicating weak to non-existent linear relationships between landscape covariates and observed thermal sensitivity."

7. As mentioned above, plotting data from the two study watersheds together to assess apparent changes in the sensitivity metrics across elevation and other physical variables may be problematic given the inherent differences in settings. Essentially all of the apparent patterns shown in Fig 1 and 3 would not exist if either watershed dataset was plotted alone.

Comparing across basins can be a powerful tool and is a common practice in hydrologic sciences, and our inclusion of differing colors for the basins was designed to acknowledge that basic-specific differences exist beyond the parameter (elevation) shown.

8. I am not sure I universally agree with this statement that leads the Discussion: 'Thermal sensitivity varies throughout the year and reflects hydrologic conditions at a given time and place within a watershed; therefore, it should not be treated as a static value.' Just because a parameter may show variability over time, does not mean the average value is not meaningful in assessing differences between sites. Daily temperature is one example, or anything else that varies diel or seasonally. I do agree there can be great value in inspecting short term to seasonal variation in air-water temp sensitivity metrics, but that is not a requirement of all studies to be useful.

We agree with the reviewer that summary metrics can be useful and informative! However, the way thermal sensitivity is typically measured, it is often conceptualized as a single, stationary value, rather than an average of multiple estimates. We believe that this is an important distinction; recognizing that a parameter shifts over time and using the average is fundamentally different from assuming a parameter is static through time. Our point here was that recognizing variability in this parameter is important (even if a mean value is eventually used), and we will work to clarify this in the manuscript.

We edited the initial paragraph of the discussion to state "Thermal sensitivity varies throughout the year and reflects hydrologic conditions at a given time and place within a watershed; therefore, it should not be conceptualized as a static value. Although summary metrics of thermal sensitivity, such as average values over the summer, can still prove useful and informative, it is essential to acknowledge the non-stationarity of the relationship between air and water temperature for a more accurate understanding of how river temperature responds to changing conditions."

9. It is typical to not assess air-water temp relations when stream temperature falls below some threshold close to freezing, as described by Ben Letcher's work and others. Was a cutoff value used here (eg 0.5 or 1 deg C?) It does not appear so for some of the winter datasets, which may not make sense conceptually. Stream and air temperature must decouple as the water starts to freeze, though perhaps these streams do not freeze (or come close)?

We did not use a cutoff value, and fully expect streams to decouple when air temperatures drop below freezing. The only stations where freezing occurs are high-elevation sites within the Wenatchee Basin. We will acknowledge this in the manuscript.

In L348-353 we state "Observed low thermal sensitivities in winter are likely due to the non-linear relationship between air and stream temperature at cold temperatures when air temperatures can dip below the water temperature-freezing limit (Mohseni et al. 1998, 1999). Air temperature covaries strongly with elevation in Pacific Northwest basins, and sites that are high in the watershed will experience a greater number of sub-freezing days, and therefore greater decoupling between air and water temperatures."

10. What do you think may drive the super low thermal sensitivities observed at some sites (eg less than 0.01?) That would seem to be possible mismatch of air and water temp data or a spring run creek totally dominated by groundwater near to the discharge source.

Numerous potential reasons for very low thermal sensitivities exist. As stated above, periods of time when air temperatures fall below freezing could cause a complete decoupling of air and water temperatures. Intense snowmelt over the spring season could result in decoupling if high temperatures melt snowpack, reducing water temperatures. Additionally, as the reviewer suggests, small tributaries dominated by groundwater could also decouple air and water temperatures.

**Minor comments**

L37: This statement could use a range of supporting citations

We will make the requested change.

We have made the requested change and included two citations to support this statement.

L41: addition of water to the stream channel impacts thermal inertia and stream temperature sensitivity, even if that water is of the same temperature as the channel.

We will include this point in the manuscript.

We have modified this sentence to state "Stream temperature is also influenced by discharge through changes to thermal inertia and residence time (Meier et al. 2003) and runoff composition where snowmelt, surface runoff, or groundwater inflow entering the stream have different temperature signatures than the stream itself (Webb and Zhang 1997, Mohseni and Stefan 1999)."

L45: 'diagnostic' tool may be better here than 'predictive' tool

We will make the requested change.

We have made the requested change.

L65: what do you mean here by 'insensitive data'? Do you mean difficulty in collecting appropriate data to calibrate/validate heat budget models or something else?

Here we are referring to data necessary to parameterize a physically based hydrologic model, such as land use and soil parameters, surface flow characteristics and input data of rainfall, evapotranspiration, and stream flow. These data generally need to be spatially distributed and may be unavailable for certain basins or regions. We will modify the sentence to include examples of necessary data.

We have modified L38-41 to state: Issues exist with process-based modelling, including intensive data and computational needs (e.g., spatially distributed land use and soil characteristics, meteorological and discharge data, etc.), limited ability to generalize across basins, and difficulty representing groundwater and subsurface flow paths (Safeeq et al. 2014).

L72: You could pull this thought out of parenthesis.

We will make this change.

We have made the requested change.

L75: 'along' river networks?

We will make this change.

We have made the requested change.

L78: It is not clear here whether you are referring specifically to statistical cluster analysis or more qualitatively to spatial groupings of streams that show similar response across the landscape

In this sentence, we were referring generally to spatial groupings of similar streams. We will modify the word "clusters" to "groupings" to avoid confusion with our formal analysis.

We switched the wording from "clusters" to "groupings".

L82: mention generally where the two experimental basins are regionally

We will add a sentence stating that the basins are located within the Pacific Northwest (western United States).

We modified L83 to explicitly state "two Pacific Northwest river basins".

L83: it is not clear what you mean here by 'characteristic regimes'

We will modify the phrasing from "characteristic" to "typical or representative" regimes.

We switched the wording from "characteristic" to "representative" regimes.

L85: perhaps add '(decreased thermal sensitivity)' after 'decoupling between air and water temperature' for clarity

We will make the requested change.

We modified this sentence to state "What are the landscape or climate factors that best predict cluster membership?"

L107: Can you clarify the subscripts for number of loggers in each basin, and also list what specific Tidbit model(s) was used?

We will make the requested change. We used HOBO TidbiT v2 (UTBI-001) water temperature data loggers, which we will include in the manuscript.

We modified L113 and L116 to state the logger models used: HOBO TidbiT v2 (UTBI-001) for water temperature and HOBO Pendant (UA-002-64) for air temperature. We also modified subscripts on L110 to explicitly state $N_{Snoqualmie}$ and $N_{Wenatchee}$ to improve clarity.

L111: please clarify these are water years in North America

We will make the requested change.

We have made the requested change.

L117: Solar shields were also used for the Tidbit loggers deployed in the water?

Yes, solar shields were fashioned to house both water and air temperature loggers.

L141: drop 'original'

We will make the requested change.

We have made the requested change.

L141: when you say 'continuous' metric what is the realized timestep of the output? Is it calculated by season or over entire datasets?

The varying coefficient linear model utilized mean daily air and water temperature for the entire time series.

We have modified L147 to state "…we employed a varying-coefficient linear model to obtain continuous, daily estimates of thermal sensitivity".

L162 and elsewhere in this section: It would be helpful to have topical sentences explaining plainly why these various calculations were done before diving into the nuts and bolts of how they were done.

This is a good point, thank you. We will make the requested changes.

We have included topical sentences for each of our methods paragraphs.

L199: Can you better explain 'the stability of clusters' concept? Again, these methods subsections tend to dive right into the details of the calculations without a clear explanation up top of why the calculations were performed. The 'why' can be gleaned, but may not be clear for readers from varied scientific backgrounds.

We will make the requested change.

We have modified L216-218 to state "To determine whether clusters assignment were stable, or preserved under a perturbed dataset similar to the original and therefore likely reflective of real differences, we conducted a bootstrapping approach where sites were sampled with replacement and then AHC was performed on the resampled data using the fpc R package (Hennig 2020)." The underlying premise behind analyzing stability is that a good clustering of the data will be reproduced over an ensemble of perturbed datasets that are nearly identical to the original data.

L220: you may want to reminder what years you are talking about.

We will make the requested change.

We have made the requested change.

L230: Are you assessing covariance by eye or statistically?

We assessed covariance informally initially, however, in our updated interannual sensitivity analysis (see above response to Reviewer 1) we will add a statistical measure of interannual covariance.

The subsection 3.2 title may be better posed not as a question

We will make the requested change.

We have changed the subsection title to "Patterns of clustering for water temperatures, air temperatures, and thermal sensitivities".

Table 1. Its probably OK, but a little odd to list Baseflow Index as a geologic variable, given the importance of groundwater levels in addition to geologic materials.

We will change the wording from "geologic" to "hydrogeologic" to clarify this.

We changed the wording from "geologic" to "hydrogeologic".

Citations

Wasserstein, R.L. & Lazar, N.A. 2016. The ASA statement on p-values: context, processes, and purpose. *The American Statistician*, 70(2): 129-133, DOI: https://doi.org/10.1080/00031305.2016.1154108.

Amrhein, V., Greenland, S., McShane, B. 2019. Scientists rise up against statistical significance. *Nature*, 567: 305-307, DOI: https://doi.org/10.1038/d41586-019-00857-9.

---

## Author Response (AR2)

**Reviewer 1 comments**

I'm generally satisfied with the responses provided by authors.

Major comment #1: Clearly present hypotheses/predictions underlying the study
Authors now state upfront that they are performing exploratory analyses and made substantial changes to Table 3 to make this clear. Authors also added a map in Supplementary Material to help understand results with regional knowledge.

Major comment #2 : Conclusions need to be better supported by analyses and results.
Authors improved in numerous places the description of results which are now described with more precision. Removing the linear fit in Table 3 also solved some of my issues with the presentation of results.

Major comment #3: Better consideration of interannual variability in thermal sensitivity
Authors did a really good job of addressing interannual variability with additional analyses presented in Appendix A.

We are grateful that the reviewer is satisfied with our revisions.

**Reviewer 3 comments**

The manuscript by McGill et al. examines spatiotemporal variability of stream thermal sensitivities for two watersheds in Washington, USA. They collected water and air temperature data from 73 sites distributed across the Snoqualmie and Wenatchee basins. Most loggers ran for seven years. The data were used in statistical models to estimate thermal sensitivities (the slope coefficient of air temperature in a linear regression model). Seasonal models were applied, as well as time-varying coefficient models. Clustering analysis was performed to group sites that shared similar thermal sensitivities. These clusters were then used to explore how thermal sensitivity varied with climate and landscape variables. They argue that thermal sensitivity showed strongest relationships with elevation, snow water equivalent, and variables representing groundwater influence. Some variables which were expected to be related to thermal sensitivity, such as percent riparian forest cover, did not vary in a systematic way. Some key conclusions made by the authors are: (1) it is essential to acknowledge the non-stationarity of the relationship between air and water temperature, (2) snow and geological characteristics shape the relationships between air and water temperatures at the study site, and (3) classifying rivers based on thermal sensitivity is a powerful tool when planning for global change.

This is my first review of this manuscript (I did not review the original submission). Overall, the manuscript is generally well written and covers a topic suitable for HESS. My general feeling is that there is considerable amount of unexplained variability in the thermal sensitivity estimates. Reporting these sorts of noisy findings can be useful, but some of the key conclusions seem more inconclusive than how they are stated. I share a couple key comments, followed by some specific feedback. I reviewed the first round of reviewer comments and replies, and I echo some of those comments and feel as though the current version could still be improved with regards to structure and results/discussion.

1) Improving structure and flow of the manuscript

As the other reviewers pointed out, the presentation of the study would be much improved if structured around some key hypotheses. The authors replied that this is an exploratory study and that they want to avoid the use of null hypothesis significance testing. I am sympathetic to those concerns (I'm glad that a p-value is nowhere in sight), but the authors could focus on framing the study around scientific (distinct from statistical) hypotheses informed by the extensive literature on spatiotemporal variability of river temperature. Even exploratory studies will have hypotheses driving the direction of the exploration. The content for doing this is already more or less in the manuscript; however, the organization is challenging to follow. For example, some hypothesized drivers are listed in the hybrid Table 3, but this table isn't referenced until page 10. In addition, there is considerable geological context provided in the discussion that could be introduced earlier so that it doesn't feel so unexpected. I suggest using a paragraph or two at the end of the introduction to better frame the study and expected outcomes from the analyses.

We attempted to clarify the structure of the manuscript, and the driving hypotheses of our study, in several ways. First, we substantially restructured the introduction. We removed a paragraph detailing differences between statistical and process-based approaches, and we now devote the second paragraph of the introduction to laying out the hypothesized impact of climate, landscape, and hydrogeologic features we examine within our paper. Table 3 (now Table 1) is referenced in the second paragraph of the introduction to better set up our scientific objectives. Additionally, we have included an introduction to the geology of the basins in the methods section, L101-106, to familiarize readers with the geologic context of the basins much earlier.

2) Challenges in linking results to underlying controls

Much of the discussion tries to link the patterns in thermal sensitivities to underlying process controls. This is difficult since the study is exploratory, focuses on correlations, and uses statistical abstracts that can be a few steps removed from the actual observational data. For example, although the coefficient estimate associated with the air temperature term in the regression models gives you an idea of how water temperature co-varies with air temperature, you lose some information about the thermal regime at that site. Since there isn't a systematic relationship between air temperature and thermal sensitivity (Figure 3a), comparing thermal sensitivities can't tell you whether a particular stream is colder or warmer than another stream during the summer (for example). However, this can be important information for diagnosing key controls on stream thermal regimes (e.g., we might expect a colder stream, in summer, to have more groundwater influence, for example). This challenge is further compounded in this study because the thermal sensitivities are then used within clustering analyses and regression trees. The authors are familiar with these sites, and which streams have colder vs warmer or stable vs dynamic thermal regimes (or what the dominant geology of the site is), but as a reader, I find it difficult to follow these connections. Understanding these connections is crucial for interpreting how the results support the conclusions of this study. I challenge the authors to rethink how this information is presented. I provide some suggestions below, but it might be helpful to show more of the actual stream temperature time series for the individual sites (maybe in the supporting information) and referring back to those data when making interpretations.

We agree that coupling thermal sensitivity with water temperature can be a useful tool to diagnose controls on stream thermal regimes, however, this effort was outside the scope of our study. The goals of this specific study were threefold: determine the spatial and temporal distribution of commonly used air-water temperature metrics across each basin, quantify the representative thermal-sensitivity regimes and determine how clusters of similar sites differ from clusters based solely on air and water temperature, and determine the landscape or climate factors that best predict thermal sensitivity cluster membership. Future work could certainly include a more thorough examination of water temperature regimes in conjunction with thermal sensitivity. Many studies exist in the Snoqualmie and Wenatchee basins looking at facets of water temperature across the year, and our thermal sensitivity insights could be coupled with empirical observations of stream temperature (Steel et al. 2016) or process-based simulations (Cristea and Burges 2010, Yan et al. 2021) as an avenue of future research. Additionally, previous studies have used bivariate clustering of the slope (thermal sensitivity) and intercept of time varying regressions (Li et al. 2016).

We do agree that air and water temperature can provide context for interpreting thermal sensitivities, and all air and water temperature data, and thermal sensitivity estimates are available to view in detail within the associated RShiny application: https://lmcgill.shinyapps.io/TimeVarying_AWC/. Users can select a basin and site from a drop-down menu to simultaneously visualize daily air and water temperature and estimates for thermal sensitivity. We encourage the reviewer to explore this feature and now reference it in the text on L186, in addition to the Data Availability statement, in the event that other readers share similar concerns. Furthermore, annual average time series for water and air temperature for every site are visualized in Figures 4 and 5.

Partly related to the above, in my own experience I have found that there can be considerable uncertainty in the estimate of the slope coefficient/thermal sensitivity. What kind of uncertainties were associated with these estimates for this study? Could these be added as uncertainty intervals to the figures?

We agree that as empirical thermal sensitivity is a statistical relationship between two time series, there are many ways to calculate its value! We would argue that every modeling framework inherently requires a series of decisions that introduce uncertainty, ranging from data selection to parameter choices, which will collectively shape the model's structure and outcomes. We attempted to address these issues through several avenues within the manuscript. First, we showcase two alternative methods in our manuscript by comparing thermal sensitivities calculated through both standard linear regression (e.g., summary metrics) and through varying coefficient linear models. Second, we attempted to clearly articulate our reasoning behind data selection in L495-513 and the uncertainties of each modeling approach in L514-532. For example, we acknowledge that selection of the bandwidth parameter for time-varying coefficient models and the averaging period (e.g., monthly, seasonal, annual) for summary metrics will impact the final calculation of thermal sensitivity. In our RShiny application we allow users to explore various bandwidth parameters to examine how shifting this value up or down impacts time varying thermal sensitivity estimates. We additionally included leave-one-out-cross-validation analyses to assess the impacts of both interannual variability and the sensitivity of relative importance estimates within the CART analysis. These sensitivity analyses are referenced within the text and results can be found in the Supplementary Material (Appendix A, Figure S7). Lastly, the uncertainties for the summary metrics can be

conceptualized by the $R^2$ value, which is an indicator of how well water temperature can be approximated by air temperature. Summaries of this parameter are shown in Table 2.

Finally, there does not appear to be any assessment of the performance of the CART model. Many of the key conclusions rely on the results of this modelling; therefore, showing the overall performance of these models seems important. These CART models have a tendency to overfit and can be sensitive to individual data points. I recommend including some sort of evaluation of the models (e.g., leave-one-out cross validation).

We have included a leave-one-out cross validation approach for the CART modeling to determine how individual points influence estimates of relative importance within our model framework (L245-248, Figure S7). In our analysis, we show that although individual points can clearly impact relative importance estimates, when all points are considered simultaneously, estimated relative importance estimates generally line up with median values from the LOOCV analysis (Figure S7). This suggests that the CART analysis consistently identifies certain variables as more influential in making predictions, and results are relatively robust to individual data points. We have also included text within Section 4.5 of the discussion that further discusses difficulties and opportunities in collecting and analyzing data on dynamic stream networks.

**Specific comments:**

L54-56: I would perhaps qualify this as '... is often the most important...' since there are conditions when solar radiation is a secondary driver of river temperature (e.g., winter periods for well-shaded reaches - see Leach et al. (2023) and maybe references within for some examples).

We completed the requested change.

L58-59: I'm not sure the Webb and Zhang (1999) or Mohseni and Stefan (1999) are the best references to support the statement that runoff composition and groundwater inflow are important influences on river temperature. The former focused on essentially point-scale heat budgets with an emphasis on energy exchanges at the air-water interface and the latter looked at air-water temperature relationships. A better reference might be Cadbury et al. (2008).

We have included the suggested reference.

L77: Typo.

We fixed the typo.

L85-86: The second objective is awkwardly worded. It seems to state whether clusters of air-water temperature correlations differ from clusters based on air and water temperature. Before reading the rest of the manuscript, this objective seemed to me to be asking the same thing. Consider rephrasing for clarity.

We have amended objective two to state "What are the representative thermal sensitivity regimes, how do they cluster on the landscape, and how do these clusters differ from clusters based on air and water temperature individually?" We hope that this clarifies the objective for readers.

L98-109: This paragraph would benefit from some specifics. For example, provide mean January and July air temperatures and give some idea of precipitation amounts. 'Wenatchee receives a greater proportion of winter precipitation as snow' - how much greater? Figure 2 provides some context, but include some summary statistics within the text, as readers unfamiliar with this region will have little context for these general statements.

We moved long-term average annual temperature and precipitation information to this section and provide additional details about individual years of data used in this analysis in L252-259.

L116: Was air temperature also logged hourly?

Yes. We clarified this point in the manuscript.

L164: This question may not make sense, as I'm not familiar with TVCMs: What window size (in days) corresponds with a bandwidth of 0.2?

The window size corresponds with 20% of the annual data, or around 73 days, with higher weight given to data closer to the point of interest.

L220-221: How were clusters with mean Jaccard coefficients between 0.5 and 0.75 treated?

We clarified this point in the manuscript. New text states "Clusters with a coefficient larger than 0.75 were considered stable, clusters with a coefficient between 0.5 and 0.75 indicate that the cluster is measuring a pattern in the data but exact site assignment may be doubtful, and clusters with a mean Jaccard coefficient of less than 0.5 were considered unstable and may not reflect a true pattern in the data (Maheu et al. 2016, Savoy et al. 2019)."

L240-242: I see you include these long-term air temperature and precipitation values here. As I noted above, I suggest moving some of these long-term values up to the study site description. Also, what do you mean by 'long-term'? Figure S1 seems to suggest 1901-2000, but this is not clear. Also, are these DayMet output? Weather station data (if so, which stations)? Please clarify where these values come from.

The long-term data mentioned in the Supplementary Material is from the NOAA National Centers for Environmental Information climate divisional time series, which has been clarified in the text. We also included the specific years included in the long-term average in the main body of the text.

L257: I thought the data only focused on total SWE, but this statement suggests a relationship with 'snowmelt events'. It's not clear when and how the analysis focused on snowmelt events. Or are the authors assuming a single snowmelt event occurring in the spring? Is this reasonable to assume? My

guess is that, given the region, these watersheds are located within a transient snow zone and snowpacks can form and melt multiple times per winter, but maybe that's an incorrect assumption?

The SWE variable used in our analysis was calculated seasonally as the difference in SWE at the start of the season and the end of the season (L151-153). We have changed the notation to ΔSWE to clarify this in our manuscript in Figure 3 and L269-270.

L254-262: I was waiting to see if there was any explanation of how these landscape variables were calculated/estimated. There is a reference to Hill et al. 2016 in Table 1, but that citation is not in reference list. I would guess that mean slope and elevation were derived from a DEM, but I have no idea where a hydraulic conductivity estimate would come from. Is this estimate for the channel bed? Surficial geology of the upslope area?

We have amended L134-141 to include more detail about covariate calculation and corrected the citation list to include Hill et al. 2016. The new text reads "Watersheds for each site were delineated and covariates describing the watersheds were derived from commonly available geostatistical products (Table 2). Covariates were divided into four broad categories: basin topography (watershed area, mean watershed elevation, average stream slope, and distance upstream), land use (percent watershed forest, riparian forest, and lake area), climate (average temperature, precipitation, and percent precipitation falling as snow), and hydrogeologic (baseflow index, hydraulic conductivity, and soil depth to bedrock). Temperature, precipitation, and percent precipitation as snow were obtained from DAYMET Daily Surface Weather data (Thornton et al. 2020) and all other landscape covariates were obtained from the Stream-Catchment (StreamCat) Database (Hill et al. 2016)." StreamCat documentation describes the development and processing of all metrics, including hydraulic conductivity and baseflow index values. For example, the hydraulic conductivity is calculated from mean lithological hydraulic conductivity (micrometers per second) content in surface or near surface geology, which we now state in Table 1. We do not currently include processing details in our manuscript, as they are easily accessible in the StreamCat database, but we would be happy to include further details in Table 2 if the editor wishes to see the change.

L282-301: Can the number of sites within each cluster be included in the text? It's done for a few clusters but including all of them would limit the need to reference back to the table.

We completed the requested change.

L302: What is meant by 'hydrogeology' here?

A previous reviewer noted that baseflow index was a hydrologic property, whereas hydraulic conductivity and soil depth to bedrock described geologic aspects of the watersheds. Hydrogeologic simply indicates variables that describe, either directly or indirectly, the distribution and movement of groundwater through soil and bedrock. We hope the changes to L134-141 clarify this point.

L317-319: I don't understand this statement, especially '... reflect aspects of river dynamics not redundant with water and air temperature.' But aren't air temperature and climate related? Also, do the results of this

study support this statement? It seems like most of the landscape variables (I assume some of these are what the authors mean by 'geology') have very weak correlations with thermal sensitivity. Even the CART analysis seems to suggest minimal explanatory power of these variables.

This statement is simply meant to indicate that thermal sensitivity reflects unique properties of river thermal regimes that are not captured by water or air temperature alone. We have clarified this statement, and the new text reads "We find that underlying geology and climate are important controls on thermal sensitivity across two Pacific Northwest river basins, and thermal sensitivities reflect aspects of river dynamics not redundant with water and air temperature."

L335: Perhaps include Kelleher et al. (2021) here. Although focused on river temperature trends, not thermal sensitivity, they make a similar key point that seasonal trends can differ from annual or just summer patterns.

We completed the requested change.

L356-357: How are the processes controlling river temperatures 'more diverse' in spring/summer than in fall/winter? I would argue all the same energy exchange processes are occurring (radiative and turbulent exchanges, advection, etc.), it is just the relative magnitudes that differ seasonally.

We have amended this sentence to use the above wording, specifically, "The greater variability of responses in spring and summer indicates that the relative magnitude of energy exchange processes controlling river temperatures are more diverse than in fall or winter".

L372: This is the first mention of glacial influence in these watersheds. How much glacial coverage is there? Which sites had upstream glaciers? Why wasn't glacial coverage included as a landscape variable?

We utilized the 2019 National Land Cover Database for the percent of the upstream watershed classified as ice/snow land cover when drafting this statement. Values are generally small within our basins, and range from 0-0.7% in the Snoqualmie basin, and 0-3.2% in the Wenatchee basin. As true glacial input is minimal within our basins, we have amended this statement to state "This is likely due to snowmelt inputs within these catchments, and points to the importance of high elevation, late-summer snowpack melt as a significant source of summer baseflow and control on water temperatures during the months of greatest heating within these watersheds."

L387: This seems to be the first time that 'geologic controls' is clarified to mean baseflow index, hydraulic conductivity and soil depth. Although this may seem obvious to some readers, I think this should be clearly stated earlier in the manuscript. Baseflow index can be influenced by factors other than groundwater (e.g., persistent, high-elevation snowpacks, glaciers, or flow regulation - especially downstream of a dam/lake, which seems to be the case for some of these sites). In addition, there are no details on where these hydraulic conductivity and soil depth estimates come from and what they represent.

Soil depth indicates the mean depth (cm) to bedrock of soils within the watershed and hydraulic conductivity is calculated from mean lithological hydraulic conductivity (micrometers per second) content in surface or near surface geology. L133-140 now includes more details on covariate selection and Table 1 includes a more detailed description of hydraulic conductivity. We assume that soil depth to bedrock and baseflow index are familiar enough to readers of HESS that we do not need to include a description, although we would be happy to include further details in Table 2 if the editor wishes to see the change.

L393: Are 'groundwater metrics' clearly important? Some of the variables that could be associated with groundwater influence often have relative variable importance values of less than 10% - that doesn't seem very important to me. Also, there is no performance evaluation of the CART model.

We have changed this wording to "hydrogeologic" to better reflect that we are referring to baseflow index, hydraulic conductivity, and soil depth in this sentence. Additionally, see our revised leave-one-out-cross-validation analysis for greater detail on the evaluation of the CART model.

L401-403: It is difficult to follow the logic here. The authors highlight that the relationships between thermal sensitivities and groundwater metrics were mixed (and in some cases they were counter-intuitive). They note uncertainty in using these metrics to capture groundwater influence, especially in mountain headwater streams. They then conclude that thermal sensitivity is a promising indicator of groundwater influence. I don't see how the results of this study support this statement.

We believe that the use of the term "a promising indicator" does imply that more work on the topic needs to be completed. We have amended L418-422 to state "The ability to use thermal sensitivity as an empirical measure of groundwater influence, therefore, shows great promise for understanding catchment processes and informing management and restoration actions at ecologically relevant scales (Snyder et al. 2015). Although our approach moves us closer to a mechanistic understanding of the relationship between thermal sensitivity and groundwater, mixed results from our analyses emphasize the need for additional targeted studies" to clarify our thinking.

L410-411: Looking at Figure 6, I can't tell that soil depth, hydraulic conductivity and baseflow index are high in streams that overlay the lower portion of the watershed. Can these be shown in a more clear and convincing way?

We have amended this line to specify sites from Clusters 1 and 4, which will provide spatial context for readers.

L404-432: A lot of geological context is suddenly provided in this section. Have the authors considered putting some of this context within the study area description? Also, are there maps to show where the measurements sites are relative to these geological features?

We amended the methods to provide geologic context earlier in the manuscript, in L101-106.

L471: Did the authors explore the sites that were located downstream of reservoirs and lakes? Could that explain some of the spatial variability observed in this study? A number of studies have highlighted that reservoirs and lakes can have a strong influence on downstream thermal regimes.

We explored this option through the inclusion of percent upstream lake area as a potential covariate.

Figure 1: Why is there a dashed line for thermal sensitivity = 0.5?

The dashed line at 0.5 is just included as a reference for easier visualization across graphs. We now state this in the figure legend.

Figure 2: Where were the SWE and precipitation data collected? How representative are these values for the entire watersheds?

We have amended the Figure 2 legend to state "Average annual discharge, SWE, and total precipitation for the outlets of the Snoqualmie and Wenatchee basins across the sampling time frame (black dashed lines) and interannual variability across the seven water years included in this analysis (gray lines). Discharge gage locations can be found in Figure 1A and 1B, and SWE and precipitation data is from DAYMET Daily Surface Weather data for the upstream watershed of each discharge gage (Thornton et al. 2020)." Discharge gages are already present in Figures 1A and 1B.

Figure 3: Please label the subplots with (A), (B), and (C), as indicated in the caption. Also, it would be interesting to see thermal sensitivity plotted against mean summer stream temperature.

We labeled the subplots with A, B, and C, and thank the reviewer for pointing out this omission. We do not show the relationship between thermal sensitivity and mean water temperature here, in order to keep the figure limited to climate covariates. However, we hope that the RShiny application and Figures 4 and 5 are illustrative of the relationship between thermal sensitivity and water temperature within our basin.

Figure 4 and 5: Can the number of sites within each cluster be shown on these figures (e.g., change the facet labels to show: 'Cluster 1 (n = XX)').

We have completed the requested change.

Table 1: What is the Hill et al. 2016 data source? It is not listed in the reference list.

We have corrected this oversight and thank the reviewer for pointing it out.

Table 2: How were the data grouped to compute these metrics? This is not clear to me. Are these simply summaries of daily mean air and water temperatures grouped by site, season and year? Or are these the means of the inter-annual thermal sensitivities estimated by the time-varying coefficient models?

We have amended the Table 2 legend to state "Air water correlation average summary metrics by basin and season. Averages are calculated as the mean value of summary metrics at all sites across each basin and season."

Figure S1: Please show precipitation anomaly in SI units.

We have completed the requested change.

References

Cadbury, S. L., Hannah, D. M., Milner, A. M., Pearson, C. P., & Brown, L. E. (2008). Stream temperature dynamics within a New Zealand glacierized river basin. River Research and Applications, 24(1), 68-89.

Kelleher, C. A., Golden, H. E., & Archfield, S. A. (2021). Monthly river temperature trends across the US confound annual changes. Environmental Research Letters, 16(10), 104006.

Leach, J. A., Kelleher, C., Kurylyk, B. L., Moore, R. D., & Neilson, B. T. (2023). A primer on stream temperature processes. Wiley Interdisciplinary Reviews: Water, e1643.

References

Cristea, N. C., and S. J. Burges. 2010. An assessment of the current and future thermal regimes of three streams located in the Wenatchee River basin, Washington State: some implications for regional river basin systems. Climatic Change 102:493–520.

Li, H., X. Deng, C. A. Dolloff, and E. P. Smith. 2016. Bivariate functional data clustering: grouping streams based on a varying coefficient model of the stream water and air temperature relationship. Environmetrics 27:15–26.

Steel, A. E., C. Sowder, and E. E. Peterson. 2016. Spatial and Temporal Variation of Water Temperature Regimes on the Snoqualmie River Network. Journal of the American Water Resources Association 52:769–787.

Yan, H., N. Sun, A. Fullerton, and M. Baerwalde. 2021. Greater vulnerability of snowmelt-fed river thermal regimes to a warming climate. Environmental Research Letters 16:054006.